# Rare X-linked variants carry predominantly male risk in autism, Tourette syndrome, and ADHD

Sheng Wang [1], Belinda Wang [1], Vanessa Drury[1], Sam Drake[1], Nawei Sun [1], Hasan Alkhairo[1], Juan Arbelaez[1], Clif Duhn [1], Tourette International Collaborative Genetics (TIC Genetics)*, Vanessa H. Bal [2], Kate Langley[3,4], Joanna Martin[3], Pieter J. Hoekstra [5,6], Andrea Dietrich[5,6], Jinchuan Xing [7], Gary A. Heiman[7], Jay A. Tischfield[7], Thomas V. Fernandez [8], Michael J. Owen[3], Michael C. O'Donovan [3], Anita Thapar [3], Matthew W. State [1] & A. Jeremy Willsey [1,9] ✉

Autism spectrum disorder (ASD), Tourette syndrome (TS), and attention-deficit/hyperactivity disorder (ADHD) display strong male sex bias, due to a combination of genetic and biological factors, as well as selective ascertainment. While the hemizygous nature of chromosome X (Chr X) in males has long been postulated as a key point of "male vulnerability", rare genetic variation on this chromosome has not been systematically characterized in large-scale whole exome sequencing studies of "idiopathic" ASD, TS, and ADHD. Here, we take advantage of informative recombinations in simplex ASD families to pinpoint risk-enriched regions on Chr X, within which rare maternally-inherited damaging variants carry substantial risk in males with ASD. We then apply a modified transmission disequilibrium test to 13,052 ASD probands and identify a novel high confidence ASD risk gene at exome-wide significance (*MAGEC3*). Finally, we observe that rare damaging variants within these risk regions carry similar effect sizes in males with TS or ADHD, further clarifying genetic mechanisms underlying male vulnerability in multiple neurodevelopmental disorders that can be exploited for systematic gene discovery.

Many neurodevelopmental disorders (NDD), including autism spectrum disorder (ASD), Tourette syndrome (TS), and attention-deficit/hyperactivity disorder (ADHD), have consistent and pronounced male sex biases[1–3]. This male sex bias remains largely unexplained. While ascertainment methods and potential diagnostic bias are likely confounds[1,4], studies that endeavored to account for these factors have nonetheless observed residual evidence for male bias[5–7].

A potential explanation for this remaining male predominance could be the so-called "female protective effect" (FPE), which may be mediated by sex-differential biological factors, such as sex hormones and/or underlying differences in development and physiology[5]. Consistent with this hypothesis, genetic studies of rare de novo and transmitted variants show an increased burden in female probands[8–10]. Similarly, common variants are overrepresented in female probands as well as in unaffected mothers[11–13].

Along these lines, differences in the canonical composition of the sex chromosomes may also contribute to female "resilience" (or male "susceptibility," depending on perspective). For example, the presence

A full list of affiliations appears at the end of the paper. *A list of authors and their affiliations appears at the end of the paper.
✉e-mail: jeremy.willsey@ucsf.edu

of a single copy of chromosome X (Chr X) in males likely results in a corresponding susceptibility to deleterious genetic abnormalities, especially within the non-pseudoautosomal region (Chr X non-PAR)[14–16].

Indeed, genetic disruptions of Chr X have long been studied in psychiatric syndromes and NDDs[17–20]. Over a hundred genes have been associated with X-linked monogenic disorders that predominantly affect males, and are often characterized by severe intellectual disability (ID), structural brain abnormalities, and/or epilepsy[19,21–23]. Interrogation of rare and severe syndromes with a highly characteristic presentation and substantial comorbidity with ASD, ADHD, epilepsies, ID, and other psychiatric and NDDs has also identified specific X-linked genetic risk factors. These include Chr X aneuploidies such as Turner syndrome and Klinefelter syndrome[18,24–28]; and disruptions of single genes on Chr X, such as *FMRP*, *MECP2*, *DMD*, and many others[21,29–35].

However, systematic, exome- and genome-wide studies of "idiopathic"[11] forms of ASD, TS, or ADHD have been less successful in identifying risk genes on Chr X[10,36–38], especially compared to the hundreds of risk genes that have been identified on the autosomes in these studies[8,39–49]. *NLGN3* and *NLGN4*, the earliest replicated genes discovered in non-syndromic ASD, identified through mapping cytogenetic abnormalities or performing parametric linkage analysis followed by targeted sequencing, map to Chr X[27,50–53]. Analyses of structural variation on Chr X have also identified putative risk regions and genes (e.g., Xp22.1 / *PTCHD1-PTCHD1AS*)[54–56]. In these cases, many risk variants are penetrant clinically only in males and have been found to be inherited from unaffected carrier mothers[51,54].

Within whole-exome sequencing (WES) case–control data, Lim et al.[57] previously observed that rare Chr X hemizygous nonsense and canonical splice-site variants were significantly enriched in male ASD probands whereas the corresponding heterozygous variants were not enriched in female probands[57]. Their findings suggested that hemizygous variants within Chr X non-PAR might carry male-specific risks and potentially explained a small proportion of male sex bias in ASD– though this was not quantified systematically and the female sample size in their study was relatively underpowered to detect such an effect. In addition, the contribution of Chr X non-PAR missense variants to ASD was not assessed and specific risk genes were not identified.

Since rare likely gene-disrupting (LGD) variants (specifically nonsense, frameshift, and canonical splice-site altering mutations) and missense variants on the autosomes carry well-replicated risks in ASD[8,10,45], it stands to reason that rare missense variants on Chr X may carry risk as well but that the signal may have been obscured by a relative lack of power. While increasingly large cohorts of patients with ASD have been sequenced (e.g., refs. 10,48,49), this question has not yet been resolved, and therefore, current estimates of the contribution of Chr X non-PAR variants to male sex bias in ASD are notably incomplete, especially as missense variants occur much more frequently than LGD variants[10]. Likewise, these questions have not been addressed in other NDDs with pronounced male bias, such as TS or ADHD, due to limited sample sizes and insufficient power. Finally, combining autosomal LGD and missense variants has been a highly successful strategy for systematic gene discovery in ASD, TS, and other neurodevelopmental and psychiatric disorders[8,19,36–38,45,58–60]. Consequently, the addition of systematic, reliable analyses of a broad range of rare coding variants mapping to Chr X non-PAR would also be expected to improve the yield of risk gene identification on Chr X.

Here, we analyzed WES data from male and female ASD probands, leveraging the family-based study design of the Simons Simplex Collection (SSC) to identify rare, maternally inherited variants on Chr X (Fig. 1). We focused on maternally inherited variants for several

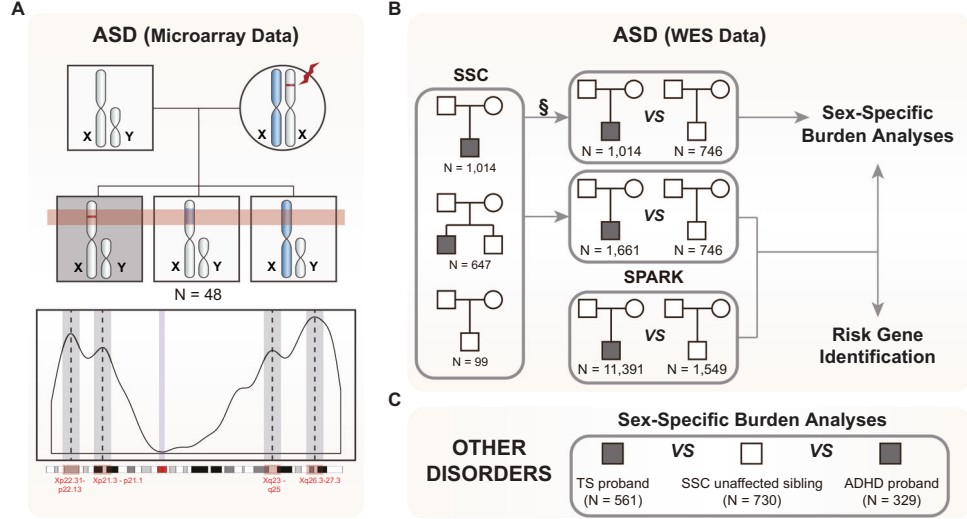

**Fig. 1 | Study schema. A** We identified risk-enriched regions (RERs) on Chromosome X (Chr X) using microarray data from 48 quintets, consisting of two unaffected parents, one male autism spectrum disorder (ASD) proband (box in dark) and two unaffected male siblings, where at least one of the unaffected siblings shares the same Chr X origin as the proband (top panel). We identified 4 peaks (RERs) within Chr X non-pseudoautosomal regions (Chr X non-PAR), encompassing a total of 149 genes (bottom panel). **B** We then utilized published whole-exome sequencing (WES) data from the Simons Simplex Collection (SSC) and SPARK ASD cohorts for (1) sex-specific burden and transmission analyses (SSC for primary burden analyses, SPARK for validation via assessment of transmission disequilibrium, combined cohort for final estimation of effect sizes based on the extent of overtransmission in probands) and (2) risk gene identification (SSC & SPARK combined). For burden analyses, (1a) we leveraged SSC siblings as controls as they are well-characterized and do not have reported psychiatric or developmental disorders. §: Chr X data is not independent for SSC families with both a male proband and one or more male siblings, and therefore, we trimmed the male probands from such families and kept the male control siblings because they are the more limiting sample set. We similarly trimmed individuals from families with multiple female children, though in this case we removed unaffected female siblings because female probands are more limiting. For transmission analyses (1b) we combined SSC and SPARK samples to investigate whether rare damaging variants are overtransmitted in male probands. In this analysis, as the untransmitted variants in each individual serve as controls, we included all individuals from the SSC cohort. For risk gene identification (2), we integrated all SSC and SPARK male samples and conducted a modified transmission disequilibrium test. **C** We extended our analyses to Tourette syndrome (TS) and attention-deficit/hyperactivity disorder (ADHD) in order to determine whether RERs carry risk in other male-biased psychiatric disorders. See also Supplementary Tables 1, 2.

reasons. First, we hypothesized that the haploid nature of Chr X in males results in a corresponding vulnerability to hemizygous variants that is not present or greatly reduced in heterozygous females[14]. Hence, in simplex families—wherein, by definition, both parents are unaffected—we further hypothesized that some unaffected mothers may be carrying deleterious Chr X variants penetrant predominantly in males, and therefore, that maternally inherited variants would be enriched in male but not female probands. This is consistent with previous work suggesting transmission of deleterious variants from "carrier" mothers to affected offspring[61]. Second, maternally inherited variants can be called with high sensitivity and specificity since identifying rare heterozygous variants in diploid mothers is routine with current state-of-the-art strategies[62]. In contrast, de novo variants on Chr X are exceedingly rare and technically challenging to call[10,62,63]; thus downstream analyses would be fraught with power issues[10].

Comparing 1014 male ASD probands to 746 male siblings from the SSC, we confirm the previously demonstrated overrepresentation of rare maternally inherited Chr X non-PAR LGD variants in male cases but do not observe evidence for the contribution of probably damaging "missense 3" (Mis3) variants alone (PolyPhen2 [HDIV] score ≥0.957 (see refs. 64,65)) or of rare damaging variants (LGD + Mis3) as a group. To better stratify risk-carrying variants, we leveraged microarray genotyping data from SSC families with multiple male children to identify specific regions within Chr X non-PAR that consistently segregated with risk—an approach conceptually similar to the newly developed stratified polygenic transmission disequilibrium test[66] (Fig. 1). Strikingly, within these regions both Mis3 variants alone as well as LGD + Mis3 variants as a group (i.e., damaging variants) showed highly significant enrichment. We then replicated this observation by demonstrating transmission disequilibrium of both LGD and Mis3 variants in males from the SPARK ASD cohort[67]. Next, we combined 1661 SSC male probands with an additional 11,391 SPARK male probands and utilized a novel modified transmission disequilibrium test to pinpoint one exome-wide significant ASD risk gene (*MAGEC3*) (Fig. 1).

Finally, we reproduced this analytic approach in TS (*N* = 561 male cases) and ADHD (*N* = 329 male cases) WES datasets (Fig. 1) and observed robust evidence for Chr X risk in males for these two strongly sex-biased psychiatric disorders, but not for epileptic encephalopathies (EE, *N* = 220 male cases) or severe undiagnosed developmental disorders (*N* = 7136 male cases), which affect males and females at a similar frequency[68–70]—suggesting that susceptibility to hemizygous damaging variants is a common mechanism in NDDs with male sex bias and that their large-scale identification offers a powerful addition to the armamentarium for systematic gene discovery in these disorders.

## Results

### Rare transmitted damaging variants are not enriched Chromosome-X-wide in ASD

We first analyzed WES data from 2058 simplex ASD families (7771 samples), including 1597 quartets and 461 trios from the SSC, representing 1975 probands and 1680 unaffected siblings[71]. For all burden analyses, we used unaffected siblings as controls. However, as proband-sibling pairs from the same family are not independent (i.e., siblings could share the same Chr X haplotype and therefore confound our analyses), we selected either one proband or one sibling from each quartet family, prioritizing male controls and female probands as these were the most limiting sample sets. After conducting extensive quality control, this resulted in 1328 ASD probands (males: 1014, females: 314) and 1557 unaffected siblings (males: 746, females: 811) (Fig. 1 and Supplementary Table 1).

We focused on rare (minor allele frequency (MAF) ≤0.1% in ExAC v0.3 and ≤0.1% within the SSC dataset), maternally inherited, hemizygous coding variants on the non-PAR of Chr X (808 genes) in male probands and SSC male control siblings. We normalized the mutation rate by the rate of rare synonymous variants in order to control for

differences in sequencing platforms and ancestry (Supplementary Table 1, Supplementary Fig. 1). We then compared the rate of hemizygous LGD variants in male ASD probands versus male control siblings and observed enrichment consistent with previous reports[49,57] (OR 1.86 [1.08–3.30], *P* = 0.028, one-sided Fisher's exact test comparing the rate of LGD variants versus synonymous variants in male probands versus unaffected male siblings; Supplementary Table 3). We did not observe enrichment of Mis3 variants alone (OR 0.94 [0.83–1.08], *P* = 0.78) or of all damaging variants (LGD + Mis3) when analyzed together (OR 0.97 [0.86–1.11], *P* = 0.65)–again, consistent with previous work[49,57]. We also did not observe significant enrichment in female ASD probands versus female control siblings for any of these variant classes (Supplementary Table 3).

### Rare transmitted damaging variants are enriched within specific regions of Chromosome X in males with ASD

Given replicated evidence for the contribution of transmitted LGD mutations on Chr X, but not for missense or damaging mutations as a group, we reasoned that strategies similar to those employed on the autosomes to stratify risk alleles, for example restricting to constrained genes, might improve signal detection and enhance gene discovery. However, constraint metrics are estimated based on selection pressures for a diploid genome, which may be different for Chr X. Consequently we investigated whether restricting the search space to Chr X regions overtransmitted from mothers to affected sons might lead to improved detection of risk alleles. This is conceptually similar to a recently developed approach leveraging common variant polygenic risk scores from the autosome to identify blocks of excess overtransmission of ASD polygenic risk (so-called stratified polygenic transmission disequilibrium test or S-pTDT)[66].

We turned to microarray genotyping data from SSC families consisting of a male proband and at least two unaffected male siblings (*n* = 65, 48 of which are informative) to identify regions on Chr X non-PAR that segregated uniquely to the male proband within a given family, with the expectation that these regions would be enriched for genes carrying damaging variants (i.e., risk genes, Fig. 1A and Supplementary Fig. 2A). We putatively term these "risk-enriched regions" as RERs and denote the remainder of Chr X non-PAR as non-enriched regions or NERs. There are 149 genes within the RERs and 659 genes in NERs (see Supplementary Table 4 for RER coordinates).

Consistent with our hypothesis, within the RERs, damaging variants as a group are significantly enriched in male probands (OR 1.60 [1.15–2.24], *P* = 0.0084, one-sided Fisher's exact test comparing the rate of damaging versus synonymous variants in cases versus controls). This signal is specific to RERs, partially driven by Mis3 variants (OR 1.61 [1.16–2.26], *P* = 0.0078), absent for more common variants (MAF > 0.1%) (Fig. 2A), and remains after excluding samples from the 48 families that were used to identify the RERs (OR 1.59 [1.13–2.23], *P* = 0.011). In addition, removing the subset of SSC probands (*n* = 313) with a pathogenic de novo mutation (see "Methods") does not alter the extent of enrichment of rare damaging variants in RERs (OR 1.65 [1.15–2.38], *P* = 0.010 vs OR 1.60, *P* = 0.0084). Moreover, our model posits that females are protected from rare heterozygous mutations, and consistent with this hypothesis, rare maternally transmitted heterozygous damaging variants in female probands do not appear to be enriched within the RERs (OR 1.10 [0.70–1.74], *P* = 0.40) (Fig. 2A). Finally, we re-ran the burden analysis using RERs defined with various size parameters and observed that our results are robust to the exact definition of the regions (Supplementary Fig. 3A).

In addition, we reasoned that risk may be increased within genes highly expressed in the brain, as has been shown for autosomal genes in ASD[41]. We leveraged BrainSpan[72,73] pre- and post-natal gene expression data from the male human brain to rank genes by overall expression across the entire exome under the presumption that, regardless of genomic location, genes with a higher brain expression

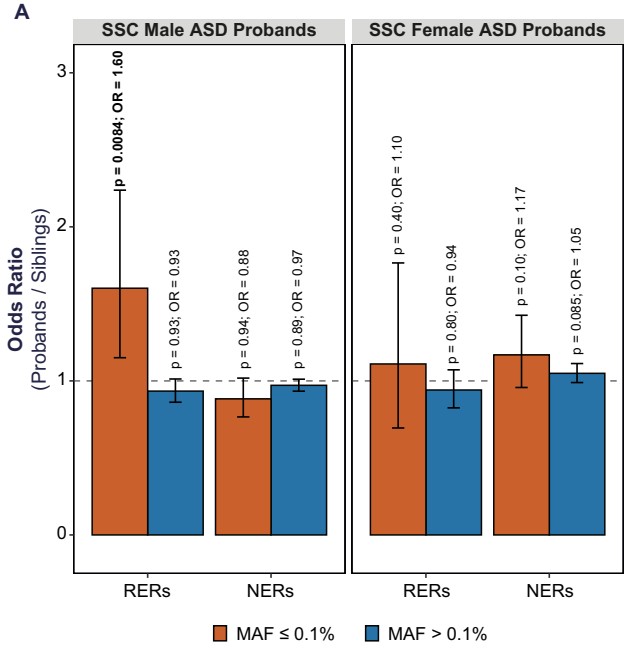

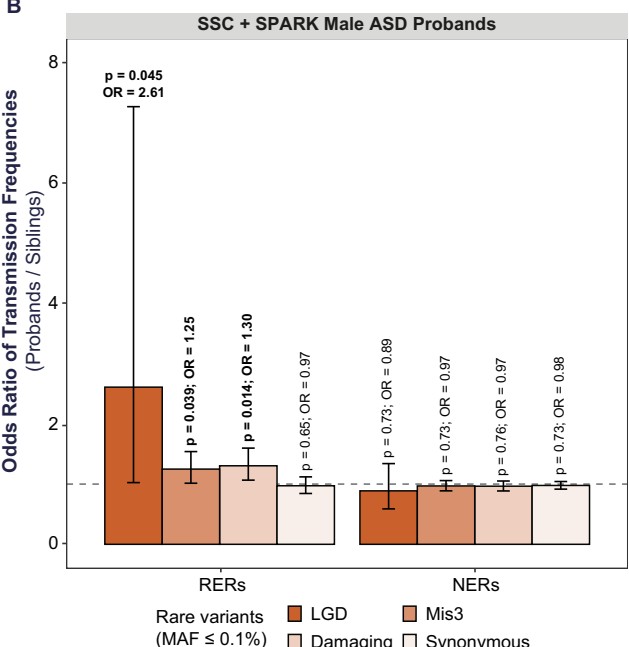

**Fig. 2 | Rare transmitted damaging variants are enriched in risk-enriched regions (RERs).** We defined RERs based on patterns of segregation in a microarray dataset from the Simons Simplex Collection (SSC) (see Fig. 1) and considered any regions outside of the four risk regions to be non-enriched regions (NERs). **A** We first compared the rate of (maternally) transmitted damaging variants in SSC probands versus SSC siblings, utilizing the rate of synonymous variants to control for potential differences in sequencing metrics and ancestries (see related Fig. 1 and Supplementary Table 1). Rare (minor allele frequency or MAF ≤ 0.1%) transmitted damaging variants are enriched in RERs in male (1014 probands versus 746 siblings) but not female probands (314 probands versus 811 siblings). Rare damaging variants in NERs are not enriched in male or female probands, nor are more

common variants (MAF > 0.1%) enriched in RERs or NERs. **B** We next orthogonally quantified the enrichment of rare damaging variants in RERs in male probands from SSC and SPARK families by comparing the transmission probabilities of rare variants (13,052 male probands versus 2295 male siblings). Separately, likely gene-disrupting (LGD) and missense 3 (Mis3; PolyPhen2 [HDIV] score ≥0.957) variants are overtransmitted. Damaging variants consist of LGD and Mis3 variants. For each bar plot, the gray horizontal line indicates odds ratio (OR) = 1, the height of the bar represents the odds ratio derived from a one-sided Fisher's exact test, and the black error bars denote the 95% confidence intervals. See also Supplementary Figs. 1–5 and Supplementary Tables 1–3.

level are more likely to be functionally important in the brain in general. We observed the same phenomenon within RERs: rare transmitted damaging variants impacting male probands are significantly enriched only in the subset of RER genes that rank among the top 25% or top 50% of brain expressed genes (top 25%, OR 2.84 [1.11–7.99], $P = 0.031$, figure not shown; top 50%, OR 2.10 [1.18–3.80], $P = 0.015$, Supplementary Fig. 2B). Again, this appears to be a male-specific effect (females: top 50%, OR 1.07 [0.45–2.50], $P = 0.53$, Supplementary Fig. 2B).

De novo autosomal variants have been reproducibly associated with decrements in IQ (particularly non-verbal IQ or NVIQ) in autism probands[8,10,45]. We therefore assessed whether rare damaging variants within RERs may similarly impact NVIQ. To do this we compared NVIQ in male probands with damaging RER variants to those without. We observed no clear difference between the two groups (mean NVIQ 87.56 versus 85.91, $P = 0.51$, two-sided $t$ test, Supplementary Fig. 2C). We also observed no clear difference between the two groups in verbal IQ (mean 83.85 versus 81.03, $P = 0.34$) or in full-scale IQ (85.43 versus 81.00, $P = 0.11$). Together, this suggests that—unlike de novo variants—RER variants do not appear to negatively impact IQ in ASD probands.

Next, we conducted several analyses to determine whether our results could be explained by a range of confounds including our normalization method or population stratification. First, we compared the unnormalized mutation rates between probands and siblings and observed the same male-specific signal for rare damaging variants (rate ratio 1.36 [1.03–1.80], $P = 0.033$, one-sided Poisson test; Supplementary Fig. 4A). Subsequently, we narrowed to the subset of SSC probands and siblings with European ancestry and repeated our burden analyses in males and females. We observed a remarkably similar

and significant effect size for rare damaging variants in European male probands (OR 1.63 [1.02–2.65], $P = 0.021$; Supplementary Fig. 4B) and the same absence of signal in European females (OR 0.75 [0.35–1.55], $P = 0.84$). In addition, we validated our results in an independent dataset and with a transmission disequilibrium test (TDT). Specifically, we compared the transmission probability of rare variants within "model-compatible" SPARK families (11,391 SPARK male probands, 1549 male sibling controls; see "Methods")[67], and observed that only rare damaging variants in RERs are overtransmitted to male probands (OR 1.26 [1.00–1.60], $P = 0.050$) and that rare synonymous variants in RERs are not overtransmitted to male probands (OR 0.97 [0.83–1.15], $P = 0.63$, figure not shown). We further validated our findings using variant calls from a recent omnibus study of ASD[49]. More specifically, within 9883 male probands, we observed that maternally inherited rare LGD variants are overtransmitted in RERs compared to NERs (OR 1.25 [1.00–1.57], $P = 0.05$, one-sided Fisher's exact test comparing transmitted versus untransmitted rare LGD variants within RERs versus NERs). However, similar to ref. 57, maternally transmitted missense variants were not reported in this study[49]. Taken together, these results suggest that the enrichment of rare damaging variants within the identified RERs are not driven by normalization methods, variant calling approaches, systematic batch effects, or ancestry, and apply to ASD cohorts with different ascertainment criteria.

Finally, we combined all the "model-compatible" male samples from the SSC and SPARK cohorts (13,052 male probands, 2295 male sibling controls, Supplementary Fig. 5) and quantified the effect sizes of LGD and Mis3 variants separately (Fig. 2B). Within this large cohort, we identified significant overtransmission of both LGD (OR 2.61 [1.02–7.27], $P = 0.045$) and Mis3 variants (OR 1.25 [1.01–1.54],

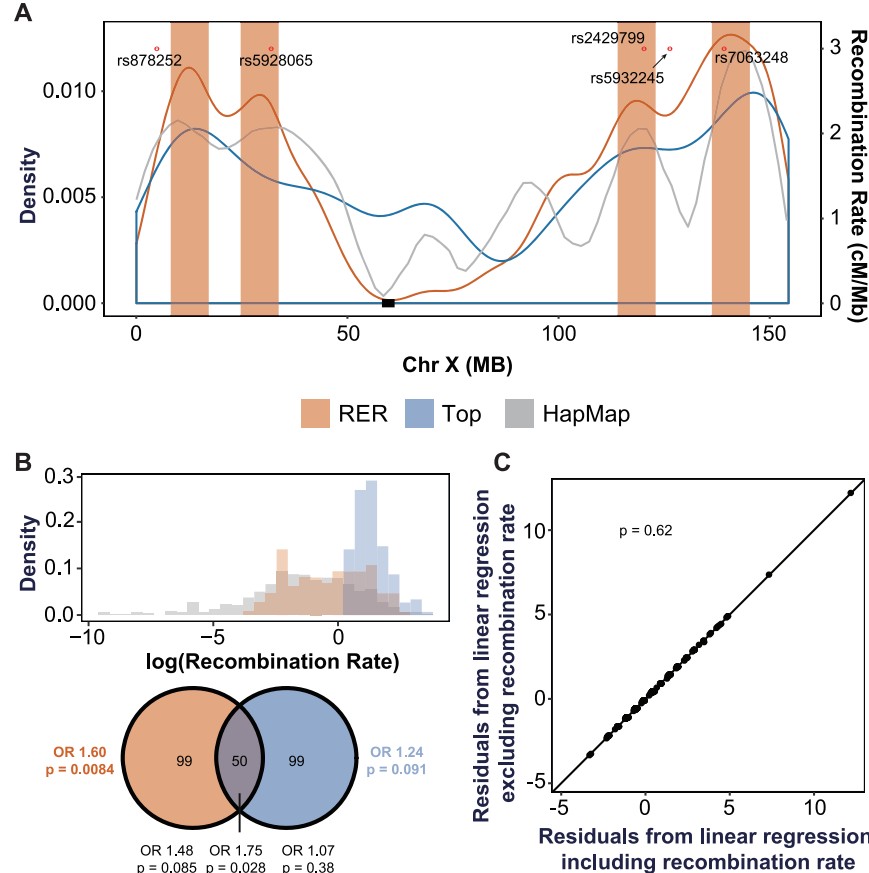

**Fig. 3 | RERs are correlated with local recombination rates. A** Density curve for risk-enriched regions (RERs) ("RER", red), and genes with the highest recombination rate ("Top", blue, generated from HapMap). The overall recombination rates from HapMap project[74] are indicated with a gray smooth line ("HapMap"). The red dots correspond to the top five SNPs identified in a Chr-X-wide association study for loci contributing to the female protective effect[86]. **B** Top panel RER genes (red) tend to have high rates of recombination compared to all Chr X non-PAR genes (gray). However, the RER genes only partially overlap the distribution of the top 149 genes based on recombination rate (blue). Bottom panel, Venn diagram depicting the overlap between the 149 genes contained within RERs and the 149 genes with the highest recombination rate. These two gene sets significantly overlap (permutation test with 100,000 iterations), but most of the risk for rare transmitted damaging variants resides within RER genes (one-sided Fisher's exact tests). **C** We compared the linear regression models for per gene counts of rare variants occurring across all of Chr X with the formula #ssc_pro.Dam - #ssc_sib.-Dam + log(recombination rate) (x-axis) and #ssc_pro.Dam - #ssc_sib.Dam (y-axis). We transformed the recombination rate to a log scale in order to make its distribution more normal. We then performed F-tests to determine whether log(recombination rate) is a significant covariate. There is no significant difference between the two models ($F = 0.253$, $P = 0.62$), suggesting that recombination rate is not a significant predictor of the per gene count of rare damaging variants in probands Chr-X-wide. OR odds ratio. See also Supplementary Fig. 6.

$P = 0.039$), as well as damaging variants as a group (OR 1.30 [1.07–1.60], $P = 0.014$). We also observe particularly strong enrichment of ultra-rare damaging variants (MAF <0.01%; Supplementary Fig. 5). Again, we did not observe signal for any of these variant types in NERs or for synonymous variants in RERs or NERs (Fig. 2B). Using these samples, we also conducted an orthogonal analysis to identify contiguous regions of overtransmission of damaging variants and observed strong overlap with the RERs used in this manuscript (Supplementary Fig. 3B), further supporting the robustness of the RER definitions.

**RERs are correlated with recombination hotspots**

Our strategy to identify RERs depends on recombination, and therefore, regions with higher recombination rates may be more likely to be identified as RERs. Indeed, the "risk" curve based on segregation in SSC families is highly similar to a curve generated solely from the recombination rates reported for Chr X as a part of the HapMap project[74] (two-sided Wilcoxon rank-sum test $P = 0.98$; Fig. 3A). To examine this overlap at the gene level, we compared the 149 genes within RERs to the 149 genes with the highest surrounding recombination rates in HapMap ("recombination" gene set). Though the 149 RER genes are enriched for genes with high recombination rate (fold enrichment 1.81,

$P = 1.0E-5$, permutation test), they occupy a much broader distribution of recombination rates (Fig. 3B). Indeed, only 50 genes of the 149 RER genes (33.6%) are present in the "recombination" gene set, suggesting that these two gene sets may have different patterns of risk. Therefore, we compared the enrichment of maternally inherited rare damaging variants from SSC male probands versus SSC male siblings (1014 probands versus 746 siblings) within the 149 "recombination" genes to the 149 RER genes. We observed significant enrichment only within the 149 RER genes (OR 1.60, $P = 0.0084$ for RER genes versus OR 1.24, $P = 0.091$ for recombination genes). We next conducted several linear regression analyses to understand whether adding recombination rate as a covariate would affect the prediction of the number of per gene rare damaging variant counts in SSC male probands from per gene variant counts in SSC male siblings (1014 probands and 746 siblings). Comparison between the residuals from both regression models did not show any significant impact of including recombination rate (F-test, $P = 0.62$ for Chr-X-wide, $P = 0.24$ for RERs, and $P = 0.86$ for NERs) (Fig. 3C and Supplementary Fig. 6A). Lastly, we ranked the genes within RERs based on recombination rate and compared the top 50% to the bottom 50%. We observed that the enrichments of rare damaging variants are similar regardless of the recombination rate ($P = 0.24$, Breslow–Day test for homogeneity of effect, Supplementary Fig. 6B).

Together, these analyses suggest that the risk in the RER gene set is not solely driven by gene-level recombination rates and that our analysis identified these RERs based on more than just recombination rate alone.

## Meta-analysis with SPARK data identifies *MAGEC3* as a high-confidence ASD risk gene

We next sought to identify specific risk genes on Chr X based on an overrepresentation of damaging variants in male probands, as has been done highly successfully for autosomal genes in ASD and other neurodevelopmental and psychiatric disorders[11,36,75]. We combined all "model-compatible" samples from the SSC and SPARK datasets, yielding a total of 13,052 male probands and 2295 controls (see Fig. 1 and Supplementary Fig. 5; "Methods"). Compared to de novo variation, gene discovery using transmitted variation is much more vulnerable to differences in ancestry between cases versus controls[59]. Therefore, we designed a modified transmission disequilibrium test (TDT), as these types of tests are more robust to population stratification because the non-transmitted variants are in effect an ancestry-matched control population[76,77]. However, one of the challenges in applying rare variant TDTs is the systematic undercalling of rare variants, which shifts the null hypothesis for transmission (detection) below 50%, thereby reducing power for detection of significant overtransmission[77]. Here, we observe even more substantial undercalling of rare maternally inherited variants on Chr X as compared to rare autosomal inherited variants (Supplementary Table 5). To account for this systematic bias, we therefore estimated null transmission probabilities for each gene based on data from control samples. Since there are a relatively small number of rare variants per gene in control samples, we estimated the local transmission probability for each gene using 3-MB bins ("Methods"). We then conducted the TDTs with these modified transmission probabilities.

Within the 149 genes in the refined RERs and using a threshold of Chr-X-wide significance ($P < 6.2E\text{-}05$ after Chr-X-wide Bonferroni correction for 808 genes), we identified a single gene—*MAGEC3*—with significant overtransmission of rare damaging variants to ASD probands versus male sibling controls ($P = 2.10E\text{-}07$, Chr-X-wide Bonferroni corrected $P = 0.00017$; Fig. 4, Table 1, and Supplementary Data 1). This gene passes exome-wide significance as well (exome-wide Bonferroni corrected $P = 0.0041$). We also observed three genes (*MAGEC1*, *SLITRK4*, and *ANOS1*) with suggestive levels of association (Fig. 4, Table 1, and Supplementary Data 1). As a control, we tested NER genes for association, and consistent with the hypothesis that these regions are depleted of risk, we did not identify any genes associated at Chr-X-wide significance within the refined NERs (Fig. 4).

## Rare damaging variants in Chr X RERs are enriched in other male-biased neurodevelopmental disorders

Prior studies have demonstrated that ASD shares genetic risk with other disorders, such as ADHD, TS, and EE[36,78–81]. However, only TS and ADHD are strongly male sex-biased[2,3,68–70]. We therefore assessed whether rare damaging variants within RERs also contribute risk to these disorders, with the hypothesis that TS and ADHD will carry risk in these regions but that EE will not. We analyzed 570 TS male probands, 332 ADHD male probands, and 223 EE male probands, using the same 1014 male SSC probands and 746 SSC male siblings as controls. For TS, we utilized 546 previously sequenced male probands[37,38] along with 24 newly sequenced male probands; for ADHD, we used 332 newly sequenced male probands; and for EE, we leveraged 223 previously sequenced male probands[81] (see also "Methods"). We did not generate exome sequencing data for the parents of the 332 ADHD probands. Therefore, for all cohorts we performed analyses using rare hemizygous variants called based on proband data alone (i.e., we did not verify transmission status and instead conducted case–control burden

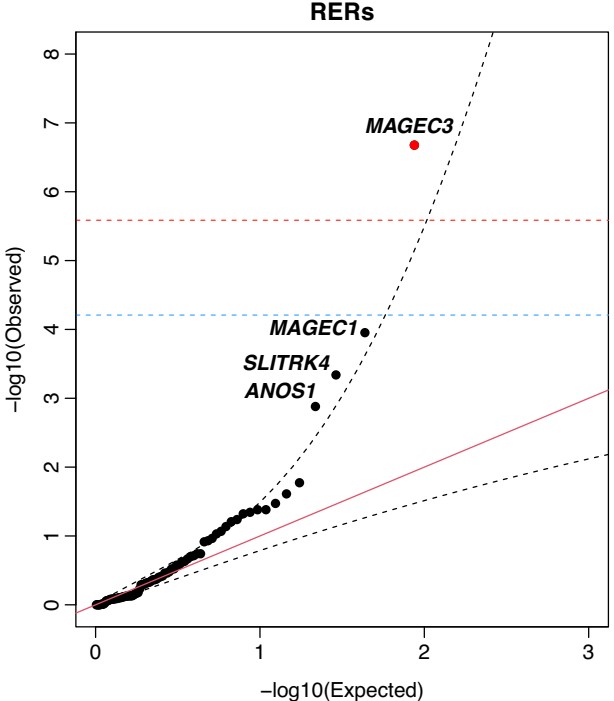

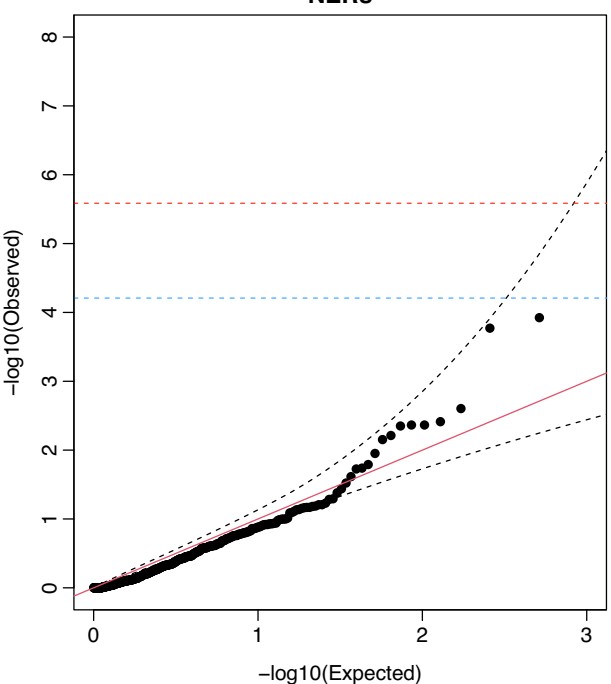

**Fig. 4 | Modified transmission disequilibrium test identifies MAGEC3 as a high-confidence ASD risk gene.** We conducted gene discovery within the risk-enriched regions (RERs) using a modified transmission disequilibrium test for rare damaging variants. As a control, we also conducted gene discovery in the non-enriched regions (NERs). In each case, we created a quantile-quantile plot comparing the distribution of *P* values to a uniform distribution (red diagonal line). Chr X non-pseudoautosomal region (Chr X non-PAR) and exome-wide significances are indicated with blue and red horizontal dashed lines, respectively. 95% confidence intervals are shown with black dashed lines. The red dot signifies the only exome-wide significant gene. Genes outside the 95% CI but not significant after correction are labeled. See also Supplementary Tables 4–5.

**Table 1 | Meta-analysis identifies *MAGEC3* as a high-confidence Autism spectrum disorder (ASD) gene**

| Gene | ID gene | P value | Bonferroni P value (Chr-X-wide) | BH FDR (Chr-X-wide) | Bonferroni P value (exome-wide) | BH FDR (exome-wide) | Brain expression level |
|---|---|---|---|---|---|---|---|
| *MAGEC3* | No | 2.10E-07 | 1.70E-04 | 1.70E-04 | 4.05E-03 | 4.05E-03 | Low |
| *MAGEC1* | No | 1.11E-04 | 0.090 | 0.032 | 1.00 | 0.77 | Low |
| *SLITRK4* | No | 4.58E-04 | 0.37 | 0.074 | 1.00 | 1.00 | High |
| *ANOS1* | No | 1.32E-03 | 1.00 | 0.18 | 1.00 | 1.00 | High |

We combined samples from the SSC and SPARK datasets and then leveraged a modified transmission disequilibrium test (TDT) to identify risk genes in ASD. To account for the systematic undercalling of rare variants, we estimated the null transmission probabilities for each gene in a window of 3MB based on data from control samples. Using this test, we identified *MAGEC3* as a high-confidence ASD gene (exome-wide Bonferroni corrected P value & Benjamini–Hochberg FDR < 0.05). We also identified three genes (*MAGEC1*, *SLITRK4*, and *ANOS1*) with suggestive levels of association. ID gene, whether a gene is a known intellectual disability (ID) gene according to ref. [91]. P value, raw p value from the modified TDT; Bonferroni P value (Chr-X-wide), TDT P value adjusted by Bonferroni correction for the 808 genes present within Chr X non-pseudoautosomal regions; BH FDR (Chr-X-wide), Benjamini–Hochberg FDR derived from the TDT P value by correcting for the 808 genes present within the Chr X non-pseudoautosomal regions; Bonferroni P value (exome-wide), TDT P value adjusted by Bonferroni correction for the 19,251 genes present exome-wide; BH FDR (Chr-X-wide), Benjamini–Hochberg FDR derived from the TDT P value by correcting for the 19,251 genes present exome-wide; Expression level: high, gene is among the top 50% of genes expressed in ASD-associated brain regions[44,98]; low, gene is among the bottom 50% of genes expressed in ASD-associated brain regions. See also Fig. 4 and Supplementary Data 1.

analyses). All samples were processed together and with the same pipeline. After quality control, we included 995 ASD probands and 730 unaffected siblings from the SSC, 561 TS probands, 329 ADHD probands, and 220 EE probands in the case–control analysis.

However, before performing these analyses, we first conducted several analyses to assess the viability of calling rare hemizygous variants in male samples without utilizing maternal data. First, using data from SSC male children only, we estimated our ability to re-identify previously called maternally inherited damaging variants. We observed a recall rate of 97.24% and a precision of 91.13%. Second, since parental whole-exome data is available for the TS probands, we analogously assessed recall rate and precision using mother-son data from the TS cohort ($n = 561$ pairs). We observed a similarly high recall rate and precision (96.90% and 91.15%, respectively). Third, after calling rare hemizygous variants in the SSC male samples, we observed that rare damaging variants are significantly enriched in SSC male cases versus controls with a remarkably consistent effect size (OR 1.45 [1.05–2.00], $P = 0.027$ for case–control versus OR 1.60, $P = 0.0084$ for maternally transmitted; see Fig. 5A versus Fig. 2A). Finally, we attempted Sanger sequencing-based confirmation of all rare coding variants identified in the ADHD samples and observed a high confirmation rate (~89–98%, see "Methods"). Together, these analyses support the viability of case–control analyses thereby enabling a more direct comparison of the effect sizes in ASD, TS, ADHD, and EE.

We next analyzed the TS and ADHD samples and, strikingly, observed that both cohorts are strongly enriched for rare hemizygous damaging variants, with effect sizes comparable to ASD (TS: OR 2.12 [1.46–3.08], $P = 0.00032$, Bonferroni adjusted P value = 0.0001; ADHD: OR 2.55 [1.60–4.08], $P = 0.00032$, adjusted P value = 0.0001; one-sided Fisher's exact test; Fig. 5A). As a negative control, we analyzed 220 male probands with EE[81], again based on the hypothesis that EE cases will not show strong enrichment for rare damaging variants within Chr X RERs due to the relative lack of sex bias in this disorder. Indeed, we do not observe a statistically significant excess in EE cases versus SSC controls (OR 1.11 [0.65–1.87], $P = 0.41$, adjusted P value = 1). Due to the relatively small size of the EE cohort, we conducted an additional validation analysis using published data[19] for a much larger cohort of individuals (7136 male cases, 8551 male controls) with severe, undiagnosed developmental disorders (DD) and a limited male sex bias (male:female = 1.4). With an analogous burden analysis, we observed that rare damaging variants in DD male cases are not significantly enriched in RER genes (OR 1.09 [0.97–1.23], $P = 0.10$). Altogether, this suggests that rare damaging variants within RERs predominantly carry risk for males in male-biased NDDs only.

ASD, TS, and ADHD are commonly comorbid[36,78–80]. Therefore, to better understand the relative effect sizes for rare hemizygous variants within Chr X RERs, we conducted a Poisson regression analysis with phenotype(s) as a covariate (Fig. 5B, C). We grouped the samples from each cohort based on comorbidity status and excluded groups with

fewer than 100 samples due to a lack of power. Again, rare damaging variants on Chr X carry comparable risk in ASD (OR 1.30 [1.00–1.70], $P = 0.051$), TS (OR 1.69 [1.22–2.34], $P = 0.004$), and ADHD (OR 1.74 [1.24–2.42], $P = 0.003$). We further explored this result by comparing rare transmitted damaging variants in RERs in SSC probands with elevated ADHD symptoms versus those without ("Methods"). While both groups were significantly enriched for damaging variants on Chr X (ASD only: OR 1.46 [1.03–2.07], $P = 0.036$; ASD with ADHD: OR 2.64 [1.48–4.73], $P = 0.0023$), the rate in ASD with elevated ADHD symptoms was significantly greater (OR 1.82 [1.03–3.20], $P = 0.041$; Supplementary Fig. 2D).

Finally, we estimated the proportion of rare damaging variants that carry risk for each disorder (Table 2)[8,37,38,45] and they range from 20–28% (ASD: 19.68% (0–46.47%), ADHD: 23.73% (0–60.16%), TS: 27.54% (0–57.68%)). We also estimated the percentage of cases in which these variants likely contribute to risk (Table 2)[8,37,38,45], yielding highly similar estimates of 2–3% (ASD: 2.36% (0–5.17%), TS: 2.74% (0–6.09%), ADHD: 2.75% (0–6.90%)). Consistent with our previous observations, in ASD probands from the SSC, the contribution of rare damaging variants in RERs varied depending on the presence or absence of elevated ADHD symptoms: these variants likely contribute risk in 5.51% of male probands with "comorbid" ADHD versus 1.98% of those without "comorbid" ADHD (Table 2 and Supplementary Fig. 2D). Altogether, these results suggest that rare hemizygous damaging variants within Chr X RERs carry broad risk for male-biased neurodevelopmental disorders and that gene discovery will be viable in larger cohorts of TS and ADHD samples.

## Discussion

Previous case–control work has shown that LGD mutations on Chr X non-PAR contribute ASD risk to males[57]. We confirm that result here, observing a similar effect size Chr-X-wide despite focusing on maternally inherited variants (Supplementary Table 3). However, in line with previous studies, we were unable to identify the enrichment of rare Mis3 variants, or damaging variants as a group, until narrowing our analysis to the 149 genes within the RERs we delineated from analyzing informative patterns of segregation in 48 simplex SSC families with multiple male children. Enrichment of damaging variants within the RERs persisted with and without normalization, does not appear to be due to population stratification, is robust across a range of RER parameters, and replicated in a large independent ASD cohort[67] as well as a recent omnibus study of ASD[49]. Together, this suggests that our results are not confounded by systematic biases in variant calling or ancestry. Likewise, we think it is unlikely these findings are driven by haplotype sharing across SSC families because we checked for cryptic relatedness before identifying RERs and conducting burden analyses, we observed consistent enrichment when excluding the 48 families used to identify the RERs, and we do not observe enrichment of rare synonymous variants or common variants within RERs. In addition to ASD, we

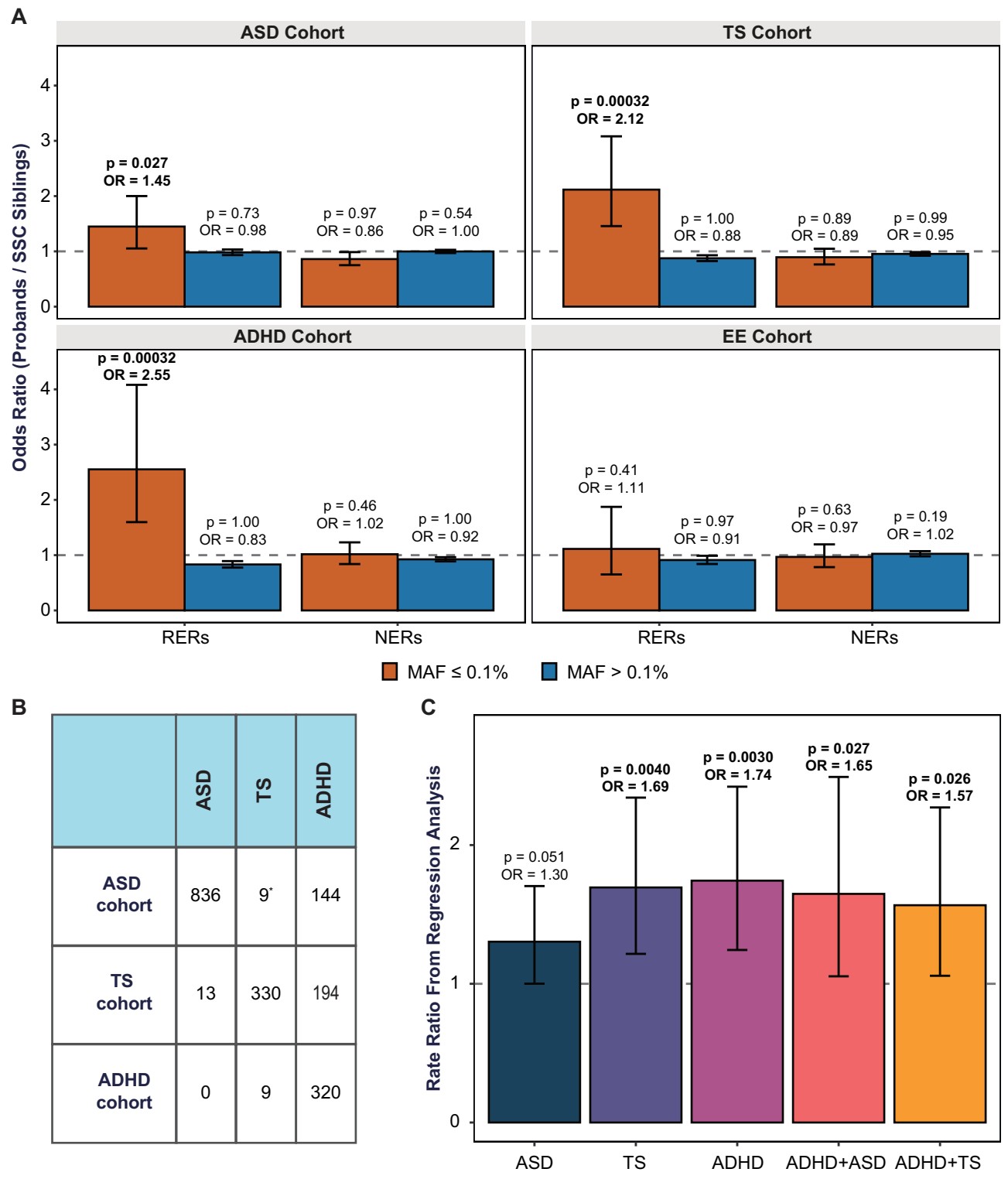

identified a clear pattern of risk for rare hemizygous damaging variants within RERs in TS and ADHD, two NDDs also with a male sex bias. However, we did not observe enrichment in EE[81] or DD[19]—two NDD cohorts with limited male sex biases. Similarly, we did not observe enrichment within female ASD patients, suggesting that damaging variants within RERs predominantly contribute to male-specific risk. That being said, damaging variants within RERs likely carry some level of risk in females[15,16]. However, we are underpowered to detect such an effect due to the small number of samples as well as the modest effect sizes observed in females. Furthermore, a female-centric approach

may identify RERs more strongly enriched for rare damaging variants in females.

Identification of the RERs relied on "informative" recombination events, therefore it is unsurprising that the RERs positively correlate with local recombination rates (Fig. 3). However, our results strongly suggest that the definition of RERs is somewhat orthogonal to recombination rate. First, stratifying genes based on recombination rate alone does not identify genes with a significant over-representation of damaging variants. Second, recombination rate is not a significant predictor of damaging variant rate in probands. Third,

**Fig. 5 | Rare damaging variants in risk-enriched regions (RERs) are also enriched in males with Tourette Syndrome (TS) or attention-deficit/hyperactivity disorder (ADHD).** We identified rare damaging variants based on proband data alone for affected males in cohorts ascertained based on ASD (n = 995), TS (n = 561), ADHD (n = 329), or epileptic encephalopathies (EE, n = 220). We also identified variants in unaffected male siblings from the SSC ASD cohort (n = 730). **A** Among these data, rare (minor allele frequency or MAF ≤0.1%) damaging variants are enriched within RERs in males from the ASD, ADHD, and TS cohorts. However, they are not enriched in males with EE, which has minimal sex bias, nor are they enriched in non-enriched regions (NERs) in any disorder. Likewise, more common variants (MAF > 0.1%) are not enriched in RERs or NERs in any disorder. We used unaffected male siblings from the SSC as controls in all comparisons. **B** Rare damaging variants appear to be more strongly enriched in male probands in the TS and ADHD cohorts, but this could be due to comorbid TS, ADHD, and/or ASD diagnoses in male

probands within each cohort. Only individuals affected by one or two disorders are listed. *In the ASD cohort probands were reported with/without Tourette/Tic disorder, whereas in the TS and ADHD cohorts, probands were reported with/without TS. **C** We therefore conducted a Poisson regression analysis with ASD, TS, and ADHD status as covariates (see (**B**) for sample sizes), the results of which further suggest that male probands with TS and/or ADHD are the most likely to have rare damaging variants in RERs. We excluded patient groups with fewer than 100 samples (e.g., ASD + TS). For each bar plot in (**A**), the gray horizontal line indicates odds ratio (OR) = 1, the height of the bar represents the odds ratio derived from a one-sided Fisher's exact test, and the black error bars denote the 95% confidence intervals. For each bar plot in (**B**) the gray horizontal line indicates rate ratio (RR) = 1, the height of the bar represents the rate ratio derived from a Poisson regression, and the black error bars denote the 95% confidence intervals. See also Table 1, Supplementary Fig. 2, and Supplementary Table 6.

**Table 2 | Rare damaging variants in RERs contribute to 2–3% of Autism spectrum disorders (ASD), Tourette Syndrome (TS), and attention-deficit/hyperactivity disorder (ADHD) diagnoses in males**

| Case–control cohort | Cases with ≥ 1 variant (%) | Controls with ≥ 1 variant (%) | Cases with contribution from ≥ 1 variant (% with 95% CI) | Variants carrying risk (% with 95% CI) |
|---|---|---|---|---|
| ASD | 10.85 | 8.49 | 2.36 (0–5.17) | 19.68 (0–46.47) |
| ASD (ADHD+) | 14.00 | 8.49 | 5.51 (0–11.68) | 35.42 (0–78.51) |
| ASD (ADHD–) | 10.47 | 8.49 | 1.98 (0–4.87) | 17.43 (0–46.15) |
| TS | 11.23 | 8.49 | 2.74 (0–6.09) | 27.54 (0–57.68) |
| ADHD | 11.25 | 8.49 | 2.75 (0–6.90) | 23.73 (0–60.16) |

We estimated the contribution of rare damaging variants using variants detected through case–control data (ASD, TS, ADHD cohorts). Based on ADHD comorbidity in the ASD cohort, we repeated the analysis for probands with ASD and ADHD (ADHD+ and for probands with ASD but no ADHD (ADHD–). Overall, we estimate that more than 2% of male cases could be explained by rare damaging variants in RERs and that more than 20% of damaging variants within RERs carry risk.

within RERs, the enrichment of rare damaging variants impacting genes with a lower recombination rate is comparable to those impacting genes with a higher recombination rate. Despite this, we cannot exclude the possibility that recombination rate intersects with ASD risk, especially because previous work in ASD has suggested that hotspots of de novo sequence and copy number variation are significantly related to recombination rate[82–84].

Due to the dependence on recombination rate and the relatively small sample size we used to identify RERs (n = 48 with informative haplotypes), we were underpowered in our definition of RERs. Because of this, we relied on a fixed window size to define RERs. Although enrichment of rare damaging variants within these regions is relatively robust to the window size (Supplementary Fig. 3A), a statistical approach with a larger number of samples may more accurately delineate RERs, thereby enriching risk and facilitating the identification of additional genes. In support of this idea, step-wise scanning of Chr X identified two additional regions of interest around the centromere that were missed in our study, likely due to the low recombination rates around these regions (Supplementary Fig. 3B). In addition, the 48 families we used are composed of at least 3 male children, and therefore, the structure of these families may have introduced bias to RER identification[85]. That being said, enrichment persisted even when removing these families from burden analyses, suggesting the RERs carry general risk for ASD. Regardless, in the future, it will be important to develop methods to identify RERs based on other family structures, including families with one child only (i.e., trios), especially as there are a large number of such families, which will likely improve power.

Interestingly, these four regions appear to co-locate with the top five SNPs from a targeted genome-wide association study (GWAS) for a common genetic locus on Chr X that may mediate the FPE in ASD[86] (Fig. 3). While the significance of this is unclear, it suggests that common variants within the RERs may also impact risk/resilience (or tag rare variants/haplotypes that impact risk). As polygenic risk scores for Chr X SNPs are generated, it will be important to address this question

more directly. Finally, even though the RERs were identified from an ASD cohort, damaging variants within these RERs also contribute risk in TS and ADHD. This might suggest that these conditions share genetic risk on Chr X, consistent with what has been observed for autosomal rare and common variation, the elevated sex ratio in probands with comorbid ASD and ADHD[36,38], and the increased rate of Chr X risk variants in SSC patients with elevated ADHD symptoms (Supplementary Fig. 2D).

Strikingly, the effect sizes estimated here parallel or exceed those observed for de novo damaging variants in ASD, TS, and ADHD[8,36,38]. However, the percentage of cases in which these variants contribute to risk is substantially lower (2–3% versus >9%)[8,36,38]. This may be due, in part, to the relatively small size of the Chr X RERs compared to the 22 autosomal chromosomes as well as the haploid nature of Chr X in males. In that light, this is a remarkably high contribution from such a small proportion of the exome.

Large-scale sequencing studies have been highly successful in idiopathic ASD—identifying hundreds of autosomal risk genes[8,10,39–49]. However, gene discovery on Chr X has lagged behind due to several reasons, including technical difficulties in calling rare variants, a general lack of signal for rare deleterious missense variants, and the rarity of de novo coding variants—all of which reduce power for gene discovery. Here, leveraging RERs, a combined dataset of 13,052 male probands and 2295 controls from the SSC and SPARK, and a modified TDT test accounting for systematic undercalling of maternally inherited Chr X rare variants, we identified a single gene, *MAGEC3*, with exome-wide significant overtransmission of rare damaging variants. *MAGEC3* belongs to the melanoma antigen (MAGE) family. Proteins from this family, which interact with E3 RING ubiquitin ligases to regulate ubiquitination, have been implicated in a wide range of disorders, including NDD[87]. This gene is also known to escape Chr X inactivation in females[88,89]. *MAGEC3* was not identified in two recent large-scale whole-exome sequencing studies that conducted a TDT on Chr X[10,49]. This may be due to several reasons. First, we designed and implemented a custom variant calling pipeline

specifically tailored to identifying rare variants on Chr X with high sensitivity and specificity. Second, Satterstrom et al. (2020) leveraged LGD and missense variants prioritized by MPC[90] whereas Zhou et al.[49] used LGD variants only. In contrast, we utilized LGD variants and missense variants prioritized by Polyphen2[64,65]. Third, we utilized a TDT that accounted for the substantial systematic undercalling we observed on Chr X. Regardless, it will be critical to replicate this association in future studies.

Despite the fact that Chr X is enriched for ID genes[19,21,91], to the best of our knowledge, *MAGEC3* has not been associated with ID. It also does not overlap with the top 23 genes identified in the recent DDD study of Chr X[19]. This is consistent with our observation that damaging variants within RERs are not associated with a change in NVIQ in autism probands−an unexpected result given that de novo damaging variants on the autosomes are strongly associated with decrements in NVIQ[8,10,45]. It will be important for future work to expand on these analyses and continue to parse the impact of damaging RER variants on NVIQ as well as the overlap in risk with ID versus ASD and other male-biased NDDs.

Notably, another member of the MAGE family, *MAGEC1* (Chr-X-wide Bonferroni corrected *P* = 0.090), was highlighted in a recent large-scale meta-analysis of schizophrenia whole-exome sequencing data[75]. Although *MAGEC3* and *MAGEC1* are adjacent to one another, we think that it is unlikely that the association of both genes is due to linkage because there is only one individual inheriting damaging variants in both *MAGEC3* and *MAGEC1* (out of 28 and 19 inherited variants, respectively) and we checked all samples for relatedness before analyses.

To account for systematic undercalling of rare variants on Chr X, we leveraged a modified TDT and transmission frequencies estimated in 3-MB windows. In the future, an increased number of control samples will enable estimates of gene-by-gene transmission frequencies thereby facilitating more precise and perhaps better-powered gene identification. Finally, novel methods for prioritizing deleterious variants (e.g., REVEL[92], MVP[93], MPC[90], metrics of constraint[94]) may improve gene discovery, though these would likely need to be calibrated to the sex chromosomes given that they have different patterns of inheritance and selection pressures.

Overall, these results raise promising hypotheses about the underlying biology of ASD, TS, and ADHD and argue for additional investigation of genetic risk on Chr X in these and other NDDs with a male sex bias, with the expectation that additional risk genes will be identified as sample sizes increase in ASD, TS, and ADHD. It will also be important to characterize the intersection of this male-specific risk factor with other contributors to risk and resilience.

## Methods
The research conducted within this manuscript complies with all relevant ethical regulations. The overall research was conducted with Institutional Review Board approval from the University of California, San Francisco.

### Whole-exome sequencing processing
All adult participants and parents of children provided written informed consent along with written or oral assent of their participating child. The Institutional Review Board or equivalent of each participating site approved the relevant study. See below for more details.

*Autism spectrum disorder (ASD)*: We obtained the whole-exome sequencing (WES) data for 2058 families from SSC, including 1597 quartets and 461 trios[71]. These samples were generated from three centers (CSHL, UW, and YALE), and were sequenced on the Illumina HiSeq 2000 sequencing platform after being captured with the NimbleGen SeqCap EZ Exome v2 array. More information about this cohort can be found in ref. 95.

*Tourette syndrome (TS)*: WES data for 546 TS trios with male probands were derived from our previous work[37,38]. In addition, we performed WES for another 24 newly recruited trios with male probands by TIC Genetics study using the xGen Exome Research Panel (IDT) capture array and sequencing on the Hiseq 4000 platform. The recruitment criteria has been described in detail previously[37,38]. All adult participants and parents of children provided written informed consent along with written or oral assent of their participating child. The Institutional Review Board of each participating site approved the study. Additional information about this cohort can be found in ref. 38.

*Attention-deficit/hyperactivity disorder (ADHD)*: We conducted whole-exome capture and sequencing for 341 ADHD male probands from the UK. Children and adolescents (aged 5−18 years) were recruited through Child and Adolescent Psychiatry or Pediatric outpatient UK clinics. ADHD diagnosis (according to DSM-III-R or DSM-IV) was confirmed using the parent version of the Child and Adolescent Psychiatric Assessment (CAPA)[96], a semi-structured diagnostic interview. Samples with ASD were excluded. Approval for the study was obtained from the North West England and Wales Multicentre Research Ethics Committees. Written informed consent to participate was obtained from all parents and from adolescents aged 16-18 years old and assent was gained from children under 16 years of age. These samples were derived from primary blood cells and captured by the xGen Exome Research Panel (IDT) capture array and sequenced on the Hiseq 4000 platform. We did not generate parental sequencing data for these samples.

*Epileptic encephalopathy (EE)*: We obtained WES data for 223 EE male probands from the Epi4K consortium. More information about these samples can be found in the published study[81].

**Quality control.** We excluded samples with unexpected relationships by using a custom script based on PLINK[97]. In addition, we excluded any samples with an inconsistent sex inferred from sex chromosome SNPs. After quality control, we obtained 1975 ASD probands from the SSC (male:female = 1,661:314), 1680 SSC siblings (male:female = 746:934), 570 male TS probands, 332 ADHD male probands, and 223 EE male probands. To avoid bias introduced by using proband-sibling pairs (i.e., shared haplotypes) from the same family in burden analyses, we further removed (1) the male ASD proband in a quartet family with two male children, as the number of male unaffected siblings is smaller than male probands and therefore more limiting; and (2) the female unaffected sibling in a quartet family with two female children, as the sample size of female probands is much smaller than female controls. This step further excluded 647 male ASD probands (1014 male ASD SSC probands remained) and 123 female SSC siblings (811 female SSC siblings remained).

In addition, we included an additional 22,416 male ASD probands and 1638 male unaffected siblings from the SPARK pilot study as well as releases 1, 2, and 3. We used Hail 0.2 (https://github.com/hail-is/hail) to liftover the variants to GRCh37. We excluded samples with inconsistent sex information inferred from the impute_sex() function. The relationship between samples was inferred with identity_by_descent() and any unexpectedly related samples were removed. We further excluded any families without maternal samples available. As a result, we included 12,441 male probands and 1621 male unaffected siblings with maternal samples in our study. Based on our model, i.e., the unaffected mother is a carrier of damaging variants penetrant mainly in males, we defined "model-compatible" families as SPARK non-multiplex families (8672 male probands and 1355 male controls) as well as multiplex families that only have multiple affected male children (i.e., both parents are unaffected and no affected female children, 2719 male probands and 194 male sibling controls). We added the latter samples based on the hypothesis that male probands in these families would be enriched for rare maternally transmitted damaging variants within Chr X RERs. Indeed, we observed significant

overtransmission of rare damaging variants (AF ≤ 0.1%) in RERs only in the SPARK model-compatible sample set (OR 1.32, $P = 0.010$) and the rate of overtransmission is not significantly different from the rate in the SPARK non-multiplex sample set alone (OR 1.07, $P = 0.33$). In total therefore, we included 11,391 male ASD probands and 1549 male unaffected siblings from the SPARK dataset.

**Family-based analysis.** We utilized GATK Best Practices for data preprocessing and variant discovery[62]. To reduce potential batch effects in variant calling, we jointly genotyped all maternal samples across the SSC ASD datasets and applied a variant quality score (VQSR) filter to extract the high quality variants. We further applied these filters to the called variants in maternal samples: (1) DP ≥ 20; (2) AB ≥ 0.3 and AB ≤ 0.7; (3) GQ ≥ 20. Instead of applying the VQSR, we conducted variant filtering separately for male probands/siblings and female probands/siblings using hard filters. More specifically, we required: (1) DP ≥ 10 for male children and DP ≥ 20 for female children; (2) AB ≥ 0.95 for male children, AB ≥ 0.3 and AB ≤ 0.7 for female children; (3) GQ ≥ 90. Again, we did not implement joint genotyping for female probands/siblings to make the analysis comparable to that in male probands/siblings. Likewise, we excluded paternally transmitted variants in female samples (i.e., the called variant should be heterozygous in mother and the offspring [like for male samples], but reference hemizygous in paternal sample). We excluded samples with an outlier number of rare variant calls (defined as mean ± 3 standard deviations).

For the SPARK samples, we could not use the same filters as above because (1) part of the VCF files from SPARK are post-filtered against VQSR; (2) variants in Chr X non-PAR in males are called diploid as well, which will affect the GQ estimation. Therefore, we adjusted our criteria as follows: (1) all variants should pass VQSR; (2) DP ≥ 20; (3) AB ≥ 0.95 for male children and AB ≥ 0.3 and AB ≤ 0.7 maternal samples [we did not analyze female children]; (4) GQ ≥ 20; (5) AF ≤ 0.1% in the maternal samples. Although these modifications potentially affect the sensitivity of the variant calling, the specificity remains the same as that in SSC data.

Within males, we labeled the variants that are maternal heterozygous and alternative hemizygous in the child as "transmitted" variants and variants that are maternal heterozygous variants but reference hemizygous in the child as "untransmitted".

**Case–control analysis.** To investigate whether the risk we observed for ASD in RERs exists in other male-biased neurodevelopmental disorders, we conducted a similar analysis using TS and ADHD male probands. We also assessed EE male probands as a negative control, given that EE does not have a strong sex bias. As maternal samples were not available for all of the probands, we conducted case–control analyses.

We generated VCF files for each male individual using the same GATK workflow as the trio data and further applied the following filters on the callset: (1) DP ≥ 10; (2) AB ≥ 0.95; (3) GQ ≥ 90; (4) AF ≤ 0.1% in our dataset as well as in ExAC v0.3 to generate the final callset. After calling, we excluded samples with an outlier number of rare variant calls (defined as mean ± 3 standard deviations). After applying these filters, we ended up with variant calls for 995 ASD SSC male probands and 730 SSC male siblings, 561 TS male probands, 329 ADHD male probands, and 220 EE male probands.

We estimated the variant calling accuracy for rare variants by comparing it with the SSC ASD trio-based (maternally transmitted) variant calls and determined 97.24% recall and 91.13% precision. We applied a similar method with the full TS trios corresponding to the TS cases utilized here and obtained 96.90% recall and 91.15% precision. For ADHD samples, we attempted to confirm all the coding variants we identified on Chr X. In total, we assessed 260 variants by Sanger sequencing. Of the 236 variants with high quality sequencing data, 232

verified (98.3% confirmation rate). The additional 24 variants were putatively confirmed, although the sequencing results were noisy. Taken together, the confirmation rate in ADHD samples is nearly 90%, even if we conservatively posited that the 24 noisy sequence validations were "false positive" calls (232 / (236 + 24) = 89.23%), which is very close to our estimation of precision using the trio dataset.

**Population stratification.** To ensure the results we observed were not driven by population stratification, we inferred the population ancestry for each dataset. We merged our dataset with genotypes from the 1000 Genomes Project and conducted the PCA analysis with the hail.pca() function. We then trained a random forest model using sklearn::RandomForestClassifier() from python3 on 1000 Genomes Project samples and predicted the ancestry of our samples. The majority of our samples were predicted to have European ancestry. When using only samples with European ancestry, rates of rare synonymous mutations are well-matched across samples from different datasets (Supplementary Fig. 1B).

### Identification of risk-enriched regions (RERs)

To identify RERs, we utilized microarray data from 65 families from the Simons Simplex Collection (SSC), each with one male proband and two or more male siblings. All the samples were genotyped on Illumina 1Mv1, 1Mv3, or Omni2.5 arrays and mapped to GRCh37 coordinates and have been reported in prior work[8]. We confirmed consistency between the reported sex and sex chromosome karyotype.

We first inferred the origins of Chr X (i.e., which maternal chromosome was inherited) for each of the children by assessing genotypes on either side of the centromere and selected the subset of families where at least one of the unaffected siblings shares the same Chr X origin as the proband ($n = 48$). Next, we identified SNPs of Chr X non-PAR that consistently segregated to the male probands. We then computed the density of 'informative recombination' events at each SNP across the 48 families, visualized the resulting chromosome-wide density plot, and identified four clear peaks. We extended 4.5 MB upstream and downstream from each peak to obtain four RERs that consistently segregate with risk (Fig. 1A; see Supplementary Data 1 for a list of the 149 genes within the RERs, and Supplementary Table 4 for the GRCh37 coordinates of the RERs).

We compared the density curve with the published HapMap dataset[74]. The entire Chr X non-PAR was split into roughly 150 bins with a 1-MB binwidth. We then calculated the average recombination rate (cM/MB) in each bin. We further normalized the average recombination rates by the sum of recombination rates of Chr X non-PAR in order to make a density distribution to compare with our dataset. For our dataset, we generated a comparable density distribution by counting the number of detected SNPs in each bin and normalizing the count by the total number of detected SNPs. We then compared the normalized values in our dataset with the normalized values from HapMap using a Wilcoxon signed rank test.

### Burden analysis
**Burden analysis for rare transmitted LGD variants on Chr X non-PAR.** To control for differences in ancestries, whole-exome capture platforms, and sequencing methods, we normalized LGD variants by synonymous variant counts with

$$fisher.text(matrix(c(\#LGD_{case}, \#Syn_{case}, \#LGD_{ctrl}, \#Syn_{ctrl}), ncol = 2),$$
$$alternative = ``greater").$$

Here we employed a one-sided test based on our model of overtransmission of rare damaging variants from unaffected mothers to affected children as well as the longstanding hypothesis that ASD cases have an elevated rate of rare damaging mutations. The odds ratio greater than 1 suggests that the odds of carrying damaging variants is

higher in cases. We further conducted the analysis using the same cutoff for allele frequency with AF ≤ 0.25% for the rare variants to make it comparable with the previous study by Lim et al.[57].

**Burden analysis for rare transmitted damaging variants.** We utilized rare synonymous variants to normalize the mutation rate in RERs and other NERs (non-enriched regions). Specifically, we performed Fisher's exact test as

$$fisher.text(matrix(c(\#Dam_{case}, \#Syn_{case}, \#Dam_{ctrl}, \#Syn_{ctrl}), ncol = 2),$$
$$alternative = ``greater")$$

to compare the rare transmitted mutation rate in male probands vs controls. Again, one-sided tests were conducted as we postulated that the mutation rate would be increased in cases. We repeated this analysis without the families used in RER identification ($n = 48$) to rule out any bias that may have been introduced by using these samples for both RER identification and burden analysis.

We performed two additional analyses as negative controls: first, we conducted the same comparisons using more common variants (ExAC MAF > 0.1%). Second, we performed the same analysis using female probands and unaffected sibling controls. We utilized rare variants that were inherited from the mother only (i.e., genotypes of a given site are heterozygous for the daughter and mother and reference hemizygous in the father). Finally, we directly compared the mutation rates in probands versus controls using a one-sided t-test and observed that the rare damaging variant rate within RERs is still elevated in male probands, thereby suggesting that enrichment of rare damaging variants within RERs is not driven by normalization via synonymous variants.

We further excluded the ASD probands carrying one or more "pathogenic" de novo mutations; more specifically, probands carrying de novo missense variants with MPC > 1, de novo LGDs, and/or de novo copy number variants, as these variants have been suggested strongly associated with ASD risk, we repeated the burden analysis and checked whether excluding these samples impacted enrichment.

Since our primary goal was to test whether the rare damaging variants within RERs are more enriched in cases, with subsequent secondary analyses focused on comparing the results to prior work or confirming that they were not due to technical or other bias, we did not correct for multiple comparisons. This procedure was followed in most analyses except for the case-control analysis of other male-biased neurodevelopmental disorders (i.e. ASD, TS, ADHD, and EE) and the gene discovery section where multiple primary tests were performed.

**Burden analysis in the SPARK only, and SSC-SPARK combined dataset.** The sample size of the SPARK dataset enables us to perform an alternative test, namely, to investigate whether rare damaging variants are more likely to be transmitted to the affected child. However, due to the undercalling of rare variants on Chr X (Supplementary Table 5), instead of using 50% transmission frequency, we compared the real transmission frequency in cases with that in controls. Specifically, we utilized 11,391 male probands and 1549 male controls from the SPARK dataset to compare the transmission frequencies of rare variants in RERs and NERs. We calculated the odds ratio with a one-sided Fisher's exact test:

$$fisher.text(matrix(c(T_{case}, T_{ctrl}, U_{case}, U_{ctrl}), ncol = 2), alternative = ``greater"),$$

where T and U are transmitted and untransmitted variant counts, respectively. We performed this analysis for rare LGD, Mis3, damaging (LGD + Mis3), and synonymous variants separately. Since this analysis is a family-based comparison, differences in population stratification and sequencing platforms are unlikely to confound the results.

We subsequently repeated this analysis after combining the SSC and SPARK dataset. Since the untransmitted variants in each individual serve as controls, we included the 647 previously removed SSC probands (that had an unaffected male sibling) to increase the statistical power in the transmission disequilibrium test (cases: 1661 SSC probands + 11,391 SPARK probands; controls: 746 SSC siblings + 1549 SPARK controls).

Moreover, to investigate whether the overtransmission of rare damaging variants in ASD probands varies in terms of allele frequencies as well as family structures. We split the samples into three categories: (1) all families; (2) model-compatible families, for which we (a) removed the families that are marked as "multiplex-multi-generational" or "multiplex-siblings-multigenerational" family type, (b) removed any families with ≥1 affected female sample; and (3) non-multiplex families (i.e., removing any families that are labeled as "multiplex" for the family type). We aggregated the transmitted variant counts for damaging variants and synonymous variants under different allele frequency cutoffs (0.01%, 0.01–0.1%, 0.1–1%, 1–10%) and performed the transmission disequilibrium test with the above formula.

**Validation of defined RERs.** To validate our definition of RERs, we performed two more analyses. First, we varied padding sizes for each peak and re-conducted the burden analysis with the SSC dataset (Supplementary Fig. 3A). The enrichment of rare damaging variants is robust as long as the extended region size is not so small that there are insufficient variants for burden analysis. Second, we utilized a sliding window (step = 20 kb) with fixed width (width = 5 MB) to identify contiguous windows within which rare damaging variants are overrepresented in ASD cases (Supplementary Fig. 3B). Here we combined the variants from SSC and SPARK, and compared transmitted and untransmitted damaging variants in cases versus controls. Because each window only had a tiny number of variants, very few windows were statistically significant, although the odds ratio did indicate a general enrichment trend in some windows. Therefore, we highlighted windows with OR > 1.2 in red regardless of the associated $P$ value (each dot denotes one window). Despite the fact that the third RER region only has a few windows, which may be because of the relatively low gene density, all four RERs (red shadow) overlap contiguous windows with suggestive enrichment (OR > 1.2). Interestingly, this analysis identified two additional regions that may also carry risk. Given our small sample size and the low recombination rate in these two regions, we hypothesize that we were unable to detect these regions due to insufficient informative recombinations (i.e., low power).

**Burden analysis regarding ADHD symptoms, non-verbal IQs, brain expression levels**

**ADHD symptoms.** We categorized our samples into two groups based on whether the sample showed elevated ADHD symptoms. We defined any probands with an ADHD t-score of CBCL age 2–5 or 6–18 subscales greater than 70 as "ASD + ADHD" (i.e., ASD comorbid with ADHD), and the remainder as "ASD - ADHD" (i.e., ASD not comorbid with ADHD). We then performed the burden analysis in the two groups (150 probands in the "ASD + ADHD" group and 830 in the "ASD - ADHD" group). Direct comparisons between ASD with or without ADHD were conducted by comparing the rare damaging mutation rate after normalizing by synonymous variants with Fisher's exact test.

**Non-verbal IQ.** We compared the distributions of reported non-verbal IQs (NVIQs) to investigate whether having rare damaging variants on RERs will affect the NVIQs, as was observed in studies of de novo variants. We compared the two groups with a two-tailed t-test.

**Brain expression level.** We attempted to determine whether genes with higher brain expression tend to have more signals as was observed in a previous study[41]. We utilized the brain expression data from male samples derived from Brainspan[72,73] and ranked the genes exome-wide according to the maximum expression level across all regions and all time points. To gain sufficient statistical power, we compared the genes with the top 50% brain expression level with the remaining genes on RERs and NERs separately.

**Regression analysis of rare damaging variants in different cohorts.** To understand the contribution of risk to different phenotypes, we performed a regression analysis using comorbidity status as a covariate. For the TS and ADHD cohorts, we used only the samples with clear records of clinical comorbidities. For the SSC cohort, we manually annotated samples as having elevated ADHD symptoms by setting a t-score cutoff of 70 from the CBCL age 2–5 or 6–18 subscales. Any probands with scores greater than the cutoff were considered to have elevated ADHD symptoms. We did not remove the small number of SSC siblings who had ADHD CBCL t-scores above the cutoff, as this results in a more conservative analysis. To improve statistical power, we excluded any groups with fewer than 100 samples. As a result, we obtained five groups of samples, including ASD only, TS only, ADHD only, ADHD + ASD, and ADHD + TS. TS comorbidity in SSC cohort is not well reported as we cannot distinguish whether the records are Tic disorder or TS. We conducted the Poisson regression analysis with the formula #dam ~ Phenotype + offset(#syn), where #dam indicates the number of rare damaging variants in RERs. We used #syn, the rare synonymous count, as an offset to control for population ancestry and batch effects introduced by sequencing platforms. We further obtained the rate ratios, 95% confidence intervals, and P values.

## Comparison of RER genes and genes with high recombination rates

We inferred the recombination rates for all genes on Chr X non-PAR using recombination rates from the HapMap dataset[74]. We then selected the same number of genes with the highest recombination rate as the "top recombination rate genes". We compared the "top recombination rate genes" with genes from RERs. Permutation testing was performed to estimate the overlap of the two sets of genes. We randomly selected the same number of genes from Chr X non-PAR and checked the overlap with the RER genes. We defined a "success" as the overlapped gene number being not less than the true overlap. We ran 100,000 permutations and obtained the p values #success/100,000. The enrichment fold was calculated by the true overlapped gene number/mean(permuted overlapped gene number). We then performed rare damaging variant burden analyses on the RER and recombination gene sets alone as well as on various intersections of these gene sets (comparing SSC probands to SSC siblings as described in Burden analysis for rare transmitted damaging variants).

## Estimation of the contribution of variants and identification of risk genes

We leveraged variants from the case–control dataset to estimate the approximate percentage of cases within which rare damaging variants in RERs contribute to diagnosis[37]. More specifically, we estimated this as the difference between the percent of probands carrying at least one rare damaging variant and the percent of unaffected SSC siblings carrying at least one rare damaging variant. We generated the 95% confidence interval by bootstrapping. Furthermore, we also estimated the percentage of rare damaging variants in RERs that carry risk by comparing the average mutation rate between cases and controls using t.test in R. We obtained the 95% confidence interval from this process directly. We applied similar methods to the variants that were

detected from trios, and the results are very close to those presented here.

## Risk gene identification with SSC and SPARK samples

We utilized rare maternally-transmitted variants with allele frequency less than 0.1% from model-compatible families to identify risk genes in the refined RERs (cases: 1661 SSC probands + 11,391 SPARK probands; controls: 746 SSC siblings + 1549 SPARK controls). Since we are lacking sufficient control samples/variants to estimate per gene transmission probabilities, we estimated the local transmission probability instead. Specifically, we split Chr X non-PAR into windows with 3-MB size (so that each RER includes three windows) and estimated the transmission probability for each window with rare damaging variants from controls, and subsequently used it as the null transmission probability for genes included in the window. For genes that are included in two adjacent bins, we used the higher transmission probability for the risk gene identification (i.e., the more conservative estimate). We further excluded windows that are with less than ten heterozygous variants in the mothers. The gene discovery analysis was performed with a one-sided binomial test for each gene:

$$binom\_test(transmitted, untransmitted + transmitted, prob, alternative = {''}greater{''}),$$

We then corrected the P values for multiple testing with the Bonferroni method at the risk regions level ($n = 149$, significant $P = 0.05/149 = 0.00034$), the Chr X non-PAR level ($n = 808$, significant $P = 0.05/808 = 6.19E\text{-}5$), and the exome-wide level ($n = 19,251$, significant $P = 0.05/19,251 = 2.60E\text{-}06$).

## Burden analysis of rare LGD variants from previous work

We applied the definition of RERs/NERs to samples from ref. 57 to do a comparative burden analysis. In their dataset, we observed 12 LGD variants in RERs from male cases and 4 LGD variants in RERs from male controls (one-sided Poisson test rate ratio 2.15, $P = 0.13$). In comparison, we found 48 and 24 LGD variants in NERs from male cases and controls, respectively (RR 1.43, $P = 0.091$). A comparison between RERs and NERs revealed a tendency of the enrichment of rare LGD variants within RERs (OR 1.49, $P = 0.37$). Missense variants were not reported in this study.

Recently, a study with larger samples published in ref. 49 enabled us to directly evaluate whether the rare LGD variants are more likely to be transmitted in the male probands. We tabulated 155 transmitted LGD variants and 119 untransmitted variants in RERs and 698 transmitted and 671 untransmitted LGD variants in NERs. Burden analysis comparing RERs and NERs showed significant overtransmission of rare LGDs in RERs (OR 1.25, $P = 0.05$). Again, missense variants were not reported in this study.

## Reporting summary

Further information on research design is available in the Nature Portfolio Reporting Summary linked to this article.

# Data availability

Summary data, including Chr X variant calls and phenotype information, is de-identified and included in the Bitbucket repository located at (https://shengw@bitbucket.org/willseylab/chrx_analysis.git). Data from SSC and SPARK data accessible upon application and approval through Simons Foundation for Autism Research Initiative SFARIbase (https://base.sfari.org). Researchers must submit a project with an IRB approval or exemption notice. TS data are accessible through four sources, including Tourette International Collaborative Genetics (TIC Genetics) Study (https://tic-genetics.org/, dbGaP: https://www.ncbi.nlm.nih.gov/projects/gap/cgi-bin/study.cgi?study_id=phs001423); Tourette Association of America International Consortium for Genetics (https://tourette.org/), BioProject: PRJNA384389; Tourette

Syndrome Genetics Southern and Eastern Europe Initiative (http://tsgenesee.mbg.duth.gr/index.html); Upsala Tourette Cohort (see https://pubmed.ncbi.nlm.nih.gov/30257206/). Newly generated TS data are deposited in the existing TIC Genetics Study dbGaP study. Researchers must get approved by the relevant Data Access Committee(s) (DAC(s)) upon providing a written Research Use Statement following dbGaP instruction. EE data are accessible by submitting an application to the Steering Committee for the Epi4K Consortium (https://www.epi4k.org/collab/). Variants for the DD project are available at https://github.com/hilarymartin/DDD_chrX.git. Variants for 1000 Genomes Project samples are publicly available at http://ftp.1000genomes.ebi.ac.uk/vol1/ftp/. ADHD data are deposited into BioProject: PRJNA1003909.

## Code availability

Code and de-identified summary data to generate the figures and tables in this manuscript are available at https://shengw@bitbucket.org/willseylab/chrx_analysis.git.

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

## Acknowledgements

We wish to thank all of the families who have participated in this study, as well as all of the individuals involved in recruitment and assessment. We greatly appreciate rapid and generous access to published data from the SSC and SPARK datasets via SFARI Base (https://base.sfari.org); the Epi4K Consortium (https://www.epgp.org/) via Daniel Lowenstein, David Goldstein, and Erin Heinzen; the Tourette International Collaborative Genetics Study (TIC Genetics, https://tic-genetics.org/); the Tourette Syndrome Genetics Southern and Eastern Europe Initiative (TSGENESEE, http://tsgenesee.mbg.duth.gr/); the Tourette Association of America International Consortium for Genetics (TAAICG, https://tourette.org/); and the Uppsala Tourette Cohort (UTC). We also thank Sarah Pyle for graphic design, Vanessa Hus Bal for expert advice, Helen R. Willsey for unwavering support and scientific feedback, and the Willsey Lab along with our extended network of colleagues for their hard work and dedication. This study was supported by grants from the National Institute of Mental Health to A.J.W. and M.W.S (R01MH115963), A.J.W. (R01NS105746), G.A.H. and J.A.T. (R01MH115958), Alyssa Rosen (R01MH115960), Donald L. Gilbert (R01MH115962), Samuel Kuperman (R01MH115961), Samuel H. Zinner (R01MH115993), Barbara J. Coffey (R01MH115959), B.W. (R25MH06048); from the Tourette Association of America to A.J.W. (Young Investigator Award); from the Human Genetics Institute of New Jersey to G.A.H. and J.A.T.; and the New Jersey Center for Tourette Syndrome and Associated Disorders (NJCTS) to G.A.H. and J.A.T. We are also grateful to the NJCTS for facilitating the inception and organization of the TIC Genetics study. This study was also supported by the Weill Institute for Neurosciences (Startup Funding to A.J.W.) and the Overlook International Foundation (to M.W.S. and A.J.W.). The ADHD study was funded by the Wellcome Trust (to A.T., M.C.O'D., and M.J.O.) and also by the MRC (A.T. and K.L.), and Action Research (A.T., M.C.O'D., and M.J.O.), with project support from Sharifah Agha, Nigel Williams, Peter Holmans, and the core lab team at the MRC Centre for Neuropsychiatric Genetics and Genomics, Cardiff University. We also thank the Tourette Association of America International Consortium for Genetics (TAAICG) and the Tourette Syndrome Genetics Southern and Eastern Europe Initiative (TSGENESEE) for their ongoing collaboration and support. The content is solely the responsibility of the authors and does not necessarily represent the official views of the NIH or other funders.

## Author contributions

Conceptualization: S.W. and A.J.W.; methodology: S.W., B.W. and A.J.W.; software: S.W. and A.J.W.; validation: S.W., V.D., S.D., N.S., H.A., J.A., C.D. and A.J.W.; formal analysis: S.W. and A.J.W.; investigation: S.W., B.W. and A.J.W.; resources: T.I.C.G., K.L., J.M., P.J.H., A.D., G.A.H., J.A.T., M.J.O., M.C.O'D., A.T., M.W.S. and A.J.W.; data curation: S.W., G.A.H., J.A.T. and A.J.W.; writing (original draft): S.W., B.W., M.W.S. and A.J.W.; writing (review and editing): S.W., B.W., N.S., J.A., C.D., T.I.C.G., V.B., J.X., G.A.H., J.A.T., T.V.F., M.C.O'D., A.T., M.W.S. and A.J.W.; visualization: S.W. and A.J.W.; supervision: A.J.W.; project administration: G.A.H., J.A.T., A.T., M.W.S. and A.J.W.; funding acquisition: K.L., G.A.H., J.A.T., M.J.O., M.C.O'D., A.T., M.W.S. and A.J.W.

## Competing interests

D.L.G. has received salary/travel/honoraria from the Tourette Association of America, the Child Neurology Society, U.S. National Vaccine Injury Compensation Program, Emalex Biosciences, PTC Therapeutics, EryDel SPA, Elsevier, and Wolters Kluwer. K.M.-V. has received financial or material research support from EU (FP7-HEALTH-2011 No. 278367, FP7-PEOPLE-2012-ITN No. 316978), DFG: GZ MU 1527/3-1 and GZ MU 1527/3-2, BMBF: 01KG1421, National Institute of Mental Health (NIMH), Tourette Gesellschaft Deutschland e.V., Else-Kröner-Fresenius-Stiftung, GW pharmaceuticals, Almirall Hermal GmbH, Abide Therapeutics, and Therapix Biosiences. She has received consultant's honoraria from Abide Therapeutics, Boehringer Ingelheim International GmbH, Bionorica Ethics GmbH, CannaMedical Pharma GmbH, Canopy Growth, Columbia Care, CTC Communications Corp., Demecan, Ethypharm GmbH, Eurox Deutschland GmbH, Global Praxis Group Limited, Lundbeck, MCI Germany, Neuraxpharm, Sanity Group, Stadapharm GmbH, Synendos Therapeutics AG, and Tilray. She is an advisory/scientific board member for Alexion, CannaMedical Pharma GmbH, Bionorica Ethics GmbH, CannaXan GmbH, Canopy Growth, Columbia Care, Ethypharm GmbH, IMC Germany, Leafly Deutschland GmbH, Neuraxpharm, Sanity Group, Stadapharm GmbH, Synendos Therapeutics AG, Syqe Medical Ltd., Therapix Biosciences Ltd., Tilray, von Mende Marketing GmbH, Wayland Group, and Zambon. She has received speaker's fees from Aphria Deutschland GmbH, Almirall, Camurus, Cogitando GmbH, Emalex, Eurox Deutschland GmbH, Ever Pharma GmbH, Meinhardt Congress GmbH, PR Berater, Spectrum Therapeutics GmbH, Takeda GmbH, Tilray, and Wayland Group. She has received royalties from Deutsches Ärzteblatt, Der Neurologie und Psychiater, Elsevier, Medizinisch Wissenschaftliche Verlagsgesellschaft Berlin, and Kohlhammer. She served as a guest editor for Frontiers in Neurology on the research topic "The neurobiology and genetics of Gilles de la Tourette syndrome: new avenues through large-scale collaborative projects", is an associate editor for "Cannabis and Cannabinoid Research" and an Editorial Board Member of "Medical Cannabis and Cannabinoids" und "MDPI-Reports" and a Scientific board member for "Zeitschrift für Allgemeinmedizin". M.C.O'D. and M.J.W. received Research Grant from Takeda Pharmaceuticals out of the scope of the present work. The remaining authors declare no competing interests.

## Additional information

[1]Department of Psychiatry and Behavioral Sciences, UCSF Weill Institute for Neurosciences, University of California, San Francisco, San Francisco, CA 94143, USA. [2]Graduate School of Applied and Professional Psychology, Rutgers University, New Brunswick, NJ, USA. [3]Centre for Neuropsychiatric Genetics and Genomics, Division of Psychological Medicine and Clinical Neurosciences, Cardiff University School of Medicine, Cardiff, Wales, UK. [4]School of Psychology, Cardiff University School of Medicine, Cardiff, Wales, UK. [5]University of Groningen, University Medical Center Groningen, Department of Child and Adolescent Psychiatry, Groningen, The Netherlands. [6]Accare Child Study Center, Groningen, The Netherlands. [7]Department of Genetics and the Human Genetics Institute of New Jersey, Rutgers, the State University of New Jersey, Piscataway, NJ, USA. [8]Yale Child Study Center and Department of Psychiatry, Yale University School of Medicine, New Haven, CT, USA. [9]Quantitative Biosciences Institute (QBI), University of California, San Francisco, San Francisco, CA 94143, USA. ✉e-mail: jeremy.willsey@ucsf.edu

## Tourette International Collaborative Genetics (TIC Genetics)

Hasan Alkhairo[1], Juan Arbelaez[1], Vanessa H. Bal[2], Yana Bromberg[10], Lawrence W. Brown[11], Xiaolong Cao[7], Keun-Ah Cheon[12], Kyungun Cheong[13], Hannyung Choi[13], Barbara J. Coffey[14], Li Deng[7], Andrea Dietrich[5,6], Sam Drake[1], Vanessa Drury[1], Clif Duhn[1], Thomas V. Fernandez[8], Carolin Fremer[15], Blanca Garcia-Delgar[16], Donald L. Gilbert[17], Danea Glover[7], Dorothy E. Grice[18], Julie Hagstrøm[19], Tammy Hedderly[20], Gary A. Heiman[7], Isobel Heyman[21], Pieter J. Hoekstra[5,6], Hyun Ju Hong[22], Chaim Huyser[23,24], Heejoo Kim[25], Young Key Kim[26], Eunjoo Kim[27], Young-Shin Kim[28], Robert A. King[8], Yun-Joo Koh[25], Sodahm Kook[29], Samuel Kuperman[30], Junghan Lee[13], Bennett L. Leventhal[28], Marcos Madruga-Garrido[31], Dararat Mingbunjerdsuk[32], Pablo Mir[33,34], Astrid Morer[16,35,36,37], Tara L. Murphy[21], Kirsten Müller-Vahl[15], Alexander Münchau[38], Cara Nasello[7], Dong Hun Oh[13], Kerstin J. Plessen[19,39], Veit Roessner[40], Eun-Young Shin[41], Dong-Ho Song[13], Jungeun Song[42], Matthew W. State[1], Nawei Sun[1], Joshua K. Thackray[7], Jay A. Tischfield[7], Frank Visscher[43], Belinda Wang[1], Sheng Wang[1], A. Jeremy Willsey[1,9]✉, Jinchuan Xing[7] & Samuel H. Zinner[44]

[10]Department of Biochemistry and Microbiology, Rutgers, New Brunswick, NJ, USA. [11]Children's Hospital of Philadelphia, Philadelphia, PA, USA. [12]Division of Child and Adolescent Psychiatry, Department of Psychiatry, Institute of Behavioral Science in Medicine, Yonsei University College of Medicine, Severance Hospital, Seoul, South Korea. [13]Yonsei University Severance Hospital, Seoul, South Korea. [14]Child and Adolescent Psychiatry, University of Miami Miller School of Medicine, Miami, FL, USA. [15]Clinic of Psychiatry, Socialpsychiatry and Psychotherapy, Hannover Medical School, Hannover, Germany. [16]Department of Child and Adolescent Psychiatry and Psychology, Institute of Neurosciences, Hospital Clinic Universitari, Barcelona, Spain. [17]Cincinnati Children's Hospital Medical Center, Cincinnati, OH, USA. [18]Icahn School of Medicine at Mount Sinai, New York, NY, USA. [19]Child and Adolescent Mental Health Center, Mental Health Services, Capital Region of Denmark, Copenhagen, Denmark. [20]Evelina London Childrens Hospital London GSTT, Kings College London Faculty of Life Sciences and Medicine, London, UK. [21]Great Ormond Street Hospital for Children, and UCL Institute of Child Health, London, UK. [22]Hallym University Sacred Heart Hospital, Anyang, South Korea. [23]Amsterdam UMC, Department of Child and Adolescent Psychiatry, Amsterdam, The Netherlands. [24]Levvel, Academic Center for Child and Adolescent Psychiatry, Amsterdam, The Netherlands. [25]Korea Institute for Children's Social Development and Rudolph Child Research Center, Seoul, South Korea. [26]Yonsei Bom Clinic, Seoul, South Korea. [27]Gangnam Severance Hospital, Yonsei University College of Medicine, Gangnam-Gu Seoul, Seoul, South Korea. [28]University of California, Department of Psychiatry, San Francisco, CA, USA. [29]Yonseisodam clinic, Seoul, Korea. [30]University of Iowa Carver College of Medicine, Iowa City, IA, USA. [31]Sección de Neuropediatría, Instituto de Biomedicina de Sevilla (IBiS), Hospital Universitario Virgen del Rocío/CSIC/Universidad de Sevilla, Seville, Spain. [32]Department of Neurology, University of Washington School of Medicine, Seattle, WA, USA. [33]Unidad de Trastornos del Movimiento. Instituto de Biomedicina de Sevilla (IBiS). Hospital Universitario Virgen del Rocío/CSIC/Universidad de Sevilla, Seville, Spain. [34]Centro de Investigación Biomédica en Red sobre Enfermedades Neurodegenerativas (CIBERNED), Madrid, Spain. [35]Institut d'Investigacions Biomediques August Pi i Sunyer (IDIBAPS), Barcelona, Spain. [36]Centro de Investigacion en Red de Salud Mental (CIBERSAM), Instituto Carlos III, Madrid, Spain. [37]University of Barcelona, Barcelona, Spain. [38]Institute of Systems Motor Science, Center of Brain, Behavior and Metabolism, University of Lübeck, Lübeck, Germany. [39]Division of Child and Adolescent Psychiatry, Department of Psychiatry, University Medical Center, University of Lausanne, Lausanne, Switzerland.

[40]Department of Child and Adolescent Psychiatry, Faculty of Medicine of the TU Dresden, Dresden, Germany. [41]Yonsei Yoo & Kim Mental Health clinic, Seoul, South Korea. [42]National Health Insurance Service Ilsan Hospital, Goyang-si, South Korea. [43]Admiraal De Ruyter Ziekenhuis, Department of Neurology, Goes, The Netherlands. [44]Department of Pediatrics, University of Washington School of Medicine, Seattle, WA, USA.

