## [Peer Review File · Nature Communications]

Rare X-Linked Variants Carry Predominantly Male Risk in Autism, Tourette Syndrome, and ADHDREVIEWER COMMENTS

Reviewer #1 (Remarks to the Author):

Wang et al. studied maternally inherited coding variants on chromosome X and their potential risk for autism, Tourette and ADHD. The authors use an initial exome dataset of 1,328 ASD probands, of which 1104 males and 314 females and compared metrics from this cohort to a cohort of 1557 unaffected siblings (746 males and 811 females).

The authors observe that likely gene disrupting variants are enriched in the male probands compared to male controls, but this effect was not found for female probands. Based on microarray data of 65 male probands and unaffected male siblings the authors identify risk-enriched regions (RERs) that are exclusively segregated to affected males. The remaining regions are denoted as non-enriched regions (NERs). The authors find that LGDs are enriched in RERs and that RER genes affected by LGDs are overrepresented in the top 50% of brain expressed genes. The authors reproduce their results in a subset of European samples as well as in an independent cohort of 11,391 male probands. The authors present several additional analyses such as associating RERs to recombination hotspots, a meta-analysis identifying a novel candidate gene, and enrichment of inherited variants for other neurodevelopmental disorders.

The manuscript addresses an interesting and exciting topic and was generally well written and relatively easy to follow. The authors to some degree try to reproduce existing findings in other cohorts and add some interesting analyses on top of this. I found the results of the authors not overly convincing mostly due to the lack of underlying data and concerns about the statistical rigor.

- Although the manuscript contains many interesting analysis and results, there is very little supporting data that accompanies that manuscript that allows for a critical review. The manuscript does not contain the variants, or genes that are the basis for all of the analyses. Overall results are described at a very high level, mostly providing summary statistics and figures without any of the actual underlying numbers or data. Without this it is impossible to verify the validity of the analyses and conclusions in the manuscript.

- It is unclear what the diagnostic status is of the probands in the cohorts that are used. I would imagine that about 30% already has a diagnosis by a de novo mutation. Are such samples included? If so, the authors should evaluate the results on the undiagnosed set of male probands. I wonder what conclusions we should draw from finding enrichment of inherited chrX variants in males already diagnosed with a pathogenic de novo variant?

- It is unclear how robust the identification of RER is (as the sample size seems relatively small). A statistical approach to identifying such regions would be more appropriate. Now regions were selected based on visual inspection and extended by (arbitrary) 4.5 Mb. Please also include the exact regions, and evidence for the regions as a supplement.

- The result of the authors that damaging variants are more inherited in RERs seems to me to be a bias from the selection of the RERs. By selecting regions that are inherited only in affected males any inherited LGDs will by definition become enriched in these regions.

- I'm concerned about the strength of the statistical evidence. Most of the significant results that the authors present barely reach significance. In addition, I note that the authors perform only single sided testing, whereas two-sided testing would be appropriate. In addition, the authors do not perform any multiple testing corrections when testing for example for multiple different categories of variants (missense, LGDs or recombination genes, overlap, RER genes). (Except for the TDT where authors perform bonferroni correction).

- I note that strangely enough when the authors reproduce their results in a much larger SPARK cohort, the statistical significance does not really increase. This is not what I would expect. Can the authors provide some kind of explanation for this?

Vice versa, for example, in the ADHD cohort. Although the authors cannot verify the transmissions status (thereby diluting their dataset) and the cohort is much smaller than the ASD cohort, the author still find significant enrichment. In the ADHD cohort the enrichment could have been purely driven by de novo variants on chrX? Can the authors exclude this?

- The authors at several points draw a conclusion from obtaining non-significant results after testing which is invalid. For example the authors conclude that RER risk variants are not associated with NVIQ based on a non-significant p-value.

- The authors often test for effects in females, and from obtaining non-significant results claim support for a male-specific effect. However, the cohort of affected females is considerably smaller than for males and therefore relatively underpowered compared to the male cohort. Given the marginal statistical significance in males it is unlikely that significance can be obtained in the females even if the effect sizes are identical.

Minor comments:

- The brain expressed results, making use of the 50% cutoff is not that convincing. The results would be much more convincing if RER genes would be enriched in the top 10% or 5% expressed genes. What brain tissue was used here? Why do the authors compare to the full exome, rather than the NER genes, or NER genes with inherited LGDs? I did not find any information about this analysis in the methods.

- In the TDT please show the corrected p-value for MAGEC3 as well. This is more informative to the reader in assessing the robustness of the association.

- The analysis that the authors claim shows that their results in RERs is not driven by recombination is not very clear or convincing to me. The analysis is based on the 149 genes with highest surrounding recombination rate, but this does not mean that genes outside of the top 149 do not still have a high recombination rate. Again, please include the underlying data that was used for the analysis in order to allow readers to verify the results.

- The association of MAGEC3 is based upon a single statistical analysis without independent replication or other supporting evidence. The authors should therefore be careful to suggest this gene as a high confidence risk gene.

- As the authors point out there are many comorbidities in neurodevelopmental disorders. The authors take this into account by performing regression analysis taking into account comorbidity. This seems to be the right approach compared to the analysis mentioned before where the cohorts are analyzed without taking into account comorbidity status. I feel that the latter analysis should therefore be removed from the manuscript. Please also provide a supplementary sample overview with the phenotypes for the samples that were used.

Reviewer #2 (Remarks to the Author):

In this thoughtful and extensive set of analyses the authors probe the role of X-chromosome variation in male-biased neurodevelopmental disorders. In short, they used family-based ASD data to identify risk enriched regions in the X chromosome that roughly correspond to recombination hotspots. They observed that maternally inherited rare “damaging” variants within these regions are enriched in male ASD cases. Furthermore, these variants are also enriched in males with Tourette’s Syndrome and ADHD (common comorbidities that are similarly male biased) but not in males with epileptic encephalopathies (a common comorbidity that is not male biased). Overall, this adds important insights into the influence of X-chromosome variation on disease and the likely etiologic overlap between several

neurodevelopmental conditions. A few points deserve consideration, and may further improve this work.

1) Line 171 and Figure S2: Odds Ratios are presented but their calculation and interpretation in words could use some clarification. The odds of [what?] are higher in male probands. The concept of enrichment is raised, but it could help the reader if the Odds Ratios throughout the manuscript were explicitly clarified in this way.

2) It may help to note the difficulties with variant pathogenicity assessments, especially regarding the lack of a gold standard. The authors should not have to redo the analyses, but a threshold from an Ensemble method such as REVEL may have been a better choice for defining “damaging” variants. <https://pubmed.ncbi.nlm.nih.gov/32843488/> A brief comment on this topic could further improve the manuscript.

3) Lines 185 to 194: This is an interesting idea and there are exiting directions that it could take. However, the analysis as currently described does not warrant the conclusion offered. If the authors are interested in RER risk variants as a potential cause of low NVIQ, then they should probably not look among probands only (ASD cases). NVIQ may be a partial determinant of proband identification. Thus, in this setting, conditioning on ASD status is conditioning an intermediate variable, or a putative effect of exposure, or a proxy for the outcome under study, or (more simply) a likely collider. This would be expected to generate bias. Thus, it probably makes more sense to look for these associations in all participants, or even better, in another population-based dataset where inclusion was not influenced by ASD status. Interestingly, the presence of an NVIQ-VIQ discrepancy may define an important ASD endophenotype or identify a general category of aberrant neurodevelopment that could lead to several diagnoses <https://pubmed.ncbi.nlm.nih.gov/24450323/> The authors could consider exploring the enrichment of RER risk variants within this group (i.e. NVIQ > VIQ - agnostic of ASD status).

4) Line 232: Why are one sided tests used in some instances? This is rarely an appropriate choice unless it is explicitly defended in the text.

5) Line 257 Bonferroni correction is quite conservative. It would be interesting and perhaps important to see how many genes had a significant over-transmission with a less conservative approach (e.g. FDR). The trade off between type 1 and type 2 error is particularly important in discovery stage research <https://pubmed.ncbi.nlm.nih.gov/25071867/> and these results could be presented in the supplement.

6) The authors appear to have further exposed the importance of deleterious hemizygous variants in male-biased Neurodevelopmental disorders. Brief mention of recent work on hemizyosity-revealed pathogenicity <https://pubmed.ncbi.nlm.nih.gov/35994124/> and long standing medical genetic observations <https://pubmed.ncbi.nlm.nih.gov/15316978/> <https://pubmed.ncbi.nlm.nih.gov/17707844/> may further improve the discussion.

Minor Points

Note that the introduction reads like a summary of the whole manuscript (including results and discussion).

Table 1 legend: “can be explained by” is strong wording that invokes causal interpretations. This probably should be tempered (“may be explained by” could work better here).

The paragraph on line 51-55 could use citations (e.g. <https://pubmed.ncbi.nlm.nih.gov/32284538/> , <https://pubmed.ncbi.nlm.nih.gov/15316978/> , <https://pubmed.ncbi.nlm.nih.gov/35994124/>)

Reviewer #3 (Remarks to the Author):

In this well-written manuscript, the authors examined the role of inherited coding variation in Autism Spectrum Disorder. They targeted rare, maternally inherited regions and variants as a possible contributor for the male-bias in ASD. The authors presented results that were inline with the literature and they attempted to implicate new risk genes; they described a new link between the X-linked gene *MAGEC3* and ASD. They used data from ADHD/TS - traits with male-bias - to test whether the contribution of inherited variants is pertinent to male-specific risk in ASD-correlated traits, utilising DEE - a disease without sex bias - as a negative control set. The methods are appropriate and their description is adequate. The interpretation of the findings is largely well-justified, albeit with occasional extrapolations based on non-significant findings. I would suggest a more conservative wording when it comes to statements made on male-specific risk or comparisons where merely the point estimates of odds are different. I hope the authors find the specific feedback below useful to their revision.

1. It was found that the RER genes are enriched for those with a high recombination rate - although they have a broader distribution. A reference, equally sized, set composed of the 149 genes with the highest surrounding recombination rates in HapMap was used for comparison and the difference in distribution was interpreted to suggest that the two gene sets may have different patterns of risk.

All in all, It is rather difficult to argue in either direction and the statements "rare damaging variants are significantly more likely to occur within the 99 RER-only genes than within the 99 recombination-only genes" in the results and "most of the risk for rare transmitted damaging variants resides within RER genes" in Fig. 3 legend are too strong for several reasons, for instance:

- Both 'unique' sets are not significantly enriched and the signal is driven by the shared genes (at least based on p values). The p value for the RER-only genes is 0.85 and for the Recomb-only genes is 0.38; the CI for the odds are not given and point estimates of the odds on their own, when the p values are this far from the lower end, are not very informative.

- The 99 recombination-only genes are by definition a subset of non-RER. The whole premise is that non-RER damaging variants are not as enriched as RER variants. Directly comparing RER vs. Top-ranked on Recombination Rate should therefore not be considered an 'additional layer of evidence' .e.g, in "Together, this suggests that the majority of risk is present within the RER gene set".

- It would appear that the RER genes resemble the general distribution of recombination rates along chrX more than the ranked set. This diminishes the suitability of 'top-ranked' as a fair comparison set.

Would it be possible to approach this differently to account for the independence? Some naive suggestions (i) by using the recombination rate as denominator for normalisation as the authors did with synonymous variants, (ii) using a regression model with the recombination rate as a covariate or (iii) exact tests between high/low recombination while using RER/non-RER as strata (or the other way around) then testing for homogeneity of effects along the two strata.

The authors also touched upon the role of expression levels of RER-genes as a plausible covariate that could modulate the enrichment. It could be relevant to see the outcomes of a control comparison in which the enrichment is examined when the genes are stratified by brain expression, agnostic to their location in RER/non-RER.

2. IQ is yet another important covariate the the authors addressed in their analysis. For verbal IQ comparisons, the cohort was split into 'above average' and 'below average' which is an unusual approach; IQ is usually dichotomised using cut-offs that would separate the lower tail (Well below average or extremely low IQ) rather than a "50/50" split around the mean/median.

- Is there any explanation for this choice and are the outcomes comparable for 'more familiar IQ group definitions'?

- How much evidence do the authors find for the statement "the enrichment of rare damaging variants may be higher in the above average NVIQ group; OR 1.40 p = 0.10)", in light of the literature?

Here, I would use more conservative wording as a significant difference is far from noticeable; Although not given in the text, I expect the 95% confidence intervals are too large with a p value of 0.1 to justify a possibility of a higher burden in the high IQ group.

3. A very interesting part of the analysis was the comparison between TS/ADHD/DEE sets and SSC male siblings. Once again, I would push back a bit on the strength and the generalizability of the statement "All together, this suggests that rare damaging variants within RERs predominantly carry risk for males in male-biased NDDs only".

- A disease without a prominent male bias AND a considerable role for inherited variants is needed to examine the premise that inherited variants contribute to male-bias. While indeed a reasonable control when it comes to the phenotypic male-bias, DEE is a poor control when it comes to the mode of inheritance. DEE is caused predominantly by de novo variants and inherited variants are a very unusual cause of DEE unless bi-allelic.

- Analysing a very small set of DEE cases against a set of SSC siblings - on top of the low power - creates further ascertainment bias as several genes cause ASD/DEE/NDD-Epilepsy and limiting the analysis to RER not segregating in siblings enriches the control set for individuals without variants more than what would have been expected under a null model.

4. Although the argument was made in a technical context, the authors suggest the validity of case-control analyses as direct comparator of effect sizes. I wonder whether the authors could discuss briefly the reasons behind the absence of MAGEC3 from the top-hits of case-control studies that analysed similar SCC and SPARK cohorts. My impression is, it is fairly unusual for high-confidence genes to change rank too much between studies using largely overlapping sets. A little more elaboration on the biological plausibility of MAGEC3 would also be very useful. The role of the MAGE family in NDD was briefly discussed. It seems there are four paralogs of MAGE among the top hits of the analysis (MAGEC3, MAGEC1, MAGEE1, MAGED2 are among the top 10 genes). Would that be a fair expectation from the gene family size or do the authors believe there is possibly an enrichment in the whole gene set?

5. The estimated percentage of cases in which RER variants contribute to risk is found to be around 2-3%, that is about 20-30 probands in the total set analysed for rare variant enrichment which included about a thousand male probands. Is there an overlap between the families used in discovering RER (n = 65) and the analysis set for rare variant enrichment (1k)? If that is indeed the case, could the enrichment be driven solely by this subset of (discovery) cases/families? Did the authors attempt to remove these families from their downstream analysis and see if the associations/enrichment would still be significant?

6. It was clearly described that the gene level case-control comparisons underwent FWER correction but I cannot clearly appreciate if the p values from the various other comparisons for enrichment analysis in the study were corrected. A lot of p values are close to 0.05 and it would be quite relevant to highlight if FDR/FWER adjustment was used, or the justification for not using any. It would also be very informative if the authors would consider adding confidence intervals to point estimates in the manuscript and to the QQ plots in Fig. 4 (e.g., using permutations/label shuffles or just an estimate from a drawn from a beta distribution). The genomic inflation parameters are odd; e.g., that the lambda from the non-RER QQ plot is lower than the RER QQ plot although the observed p values look closer to the expected ones based on the drawn line; the 'overall' appearance of the RER plot is that of a slightly inflated one but lambda is less than 1.

7. Figure S4 suggests a trend of increasing effect sizes with lower MAF. With a fairly high MAF cut-off like 0.1%, it would be informative to comment on the actual MAF bin that seems to drive the enrichment (if these are mostly ultra-rare variants seen in one family), for example, in MAGEC3.

8. I would appreciate it if the authors elaborated a little in their discussion on the bias created by these two elements when it comes to inferring male-specific risk:

(i) the design that scans males only for risk region and, to a lesser extent,

(ii) the prioritisation strategy in families with more than one individuals.

The authors set out to find, and successfully identified, regions within Chr X non-PAR that consistently segregated with risk in males. The design was made to prioritise regions carrying variation that have a preferentially high effect size in males (not in two unaffected males) but not as much an effect in females (inherited from a phenotypically unaffected mother).

Given that these regions were selected to segregate with ASD in males, the odds of an affected male having a rare variant in these region when compared to other (non-segregating) regions in affected males or to the same regions in (male) controls would anyways have a higher prior to begin with.

The exact tests used (with 'greater' as the alternative) assume the null is an 'equal' or a 'lower' burden. These should work just fine for gene-level scans without prioritisation on regions but will not capture the actual "null" distribution if limited to, or stratified by, segregating regions. It does not necessarily come as a surprise that inherited missense variants and PTVs in these regions are significantly enriched in males. The p values for variant enrichment could be nominally significant (see FWER above) and the enrichment could be a signal driven by the underlying haplotype sharing not the rare variants.

REVIEWER COMMENTS

Reviewer #1 (Remarks to the Author):

Wang et al. studied maternally inherited coding variants on chromosome X and their potential risk for autism, Tourette and ADHD. The authors use an initial exome dataset of 1,328 ASD probands, of which 1104 males and 314 females and compared metrics from this cohort to a cohort of 1557 unaffected siblings (746 males and 811 females).

The authors observe that likely gene disrupting variants are enriched in the male probands compared to male controls, but this effect was not found for female probands. Based on microarray data of 65 male probands and unaffected male siblings the authors identify risk-enriched regions (RERs) that are exclusively segregated to affected males. The remaining regions are denoted as non-enriched regions (NERs). The authors find that LGDs are enriched in RERs and that RER genes affected by LGDs are overrepresented in the top 50% of brain expressed genes. The authors reproduce their results in a subset of European samples as well as in an independent cohort of 11,391 male probands. The authors present several additional analyses such as associating RERs to recombination hotspots, a meta-analysis identifying a novel candidate gene, and enrichment of inherited variants for other neurodevelopmental disorders.

The manuscript addresses an interesting and exciting topic and was generally well written and relatively easy to follow. The authors to some degree try to reproduce existing findings in other cohorts and add some interesting analyses on top of this. I found the results of the authors not overly convincing mostly due to the lack of underlying data and concerns about the statistical rigor.

- Although the manuscript contains many interesting analysis and results, there is very little supporting data that accompanies that manuscript that allows for a critical review. The manuscript does not contain the variants, or genes that are the basis for all of the analyses. Overall results are described at a very high level, mostly providing summary statistics and figures without any of the actual underlying numbers or data. Without this it is impossible to verify the validity of the analyses and conclusions in the manuscript.

We thank the reviewer for these kind and helpful comments. To allow readers to reproduce all the figures and tables in this study, we have now included full datasets along with the scripts on bitbucket: https://shengw@bitbucket.org/willseylab/chrx_analysis.git.

- It is unclear what the diagnostic status is of the probands in the cohorts that are used. I would imagine that about 30% already has a diagnosis by a de novo mutation. Are such samples included? If so, the authors should evaluate the results on the undiagnosed set of male probands. I wonder what conclusions we should draw from finding enrichment of inherited chrX variants in males already diagnosed with a pathogenic de novo variant?

This is a great point, thank you for your suggestion. To answer this question, we first obtained, from recent publications (PMID: 31981491, 26402605), de novo sequence and copy number variants for all of the SSC samples included in this manuscript. We considered de novo protein-truncating variants, missense variants with MPC>1, as well as deletions and

duplications as pathogenic—as all of these variants have been demonstrated to be significantly associated with ASD risk in previous work (PMID: 31981491). We then compared the odds ratio for rare maternally inherited Chr X variants in all SSC male probands (1014 samples) to the odds ratio in SSC male probands without a de novo sequence variant (701 samples). As shown below, there does not appear to be a difference in burden between the two groups, suggesting that our results are not driven by de novo sequence variants. Additionally, rare maternally inherited Chr X variants are not significantly enriched within the subgroup of SSC probands with a pathogenic de novo variant (313 samples, result not shown). We have added the former result to the main text in the section “Rare transmitted damaging variants are enriched in defined risk-enriched regions in males with ASD”.

- It is unclear how robust the identification of RER is (as the sample size seems relatively small). A statistical approach to identifying such regions would be more appropriate. Now regions were selected based on visual inspection and extended by (arbitrary) 4.5 Mb. Please also include the exact regions, and evidence for the regions as a supplement.

Thank you for your comments. While the exact regions were highlighted in the Methods section, we agree with the reviewer that the regions need to be more explicitly referenced. We therefore added these regions to a supplemental table (Table S3) and now refer to this table in the main text section where the RERs are first introduced. We also added the underlying data and script for generating the RER density curve shown in Figure 1 and Figure 3 to Bitbucket (https://shengw@bitbucket.org/willseylab/chrx_analysis.git).

We concur with the reviewer that the definition of the RERs was somewhat arbitrary. However, previously published genomic studies of ASD have similarly defined blocks for analysis (e.g., PMID: 36280734, 28540026). Either way, we agree that a statistical approach to identifying regions would be more appropriate. Unfortunately, however, given that we have

genotyping data from only 65 families with an informative makeup (male proband and multiple unaffected male siblings), we are vastly underpowered with respect to leveraging a statistical approach to identify RERs. Therefore, we performed two new analyses to validate the definition of the RERs. First, we re-ran the burden analysis using RERs with various extended region sizes in order to ensure that our results are robust to the exact definition of the regions (see (A) below). Second, we conducted an orthogonal analysis to identify RERs and assessed the overlap with the RERs used in this manuscript (see (B) below). Both of these analyses support the assertion that the RERs are robustly defined and capture true biological signals. These results, described in more detail below, have been incorporated into a new supplemental figure (Figure S4), and are mentioned in the main text.

(A) We re-ran the burden analysis using RERs with various extended region sizes and the SSC dataset (Fisher exact test comparing inherited damaging versus inherited synonymous variants in cases versus controls). The enrichment of rare damaging variants is robust as long as the extended region size is not so small that there are not enough variants for burden analysis. (B) We utilized an orthogonal sliding window analysis (step = 20kb) with fixed width (width = 5 MB) to determine whether rare damaging variants are over transmitted in ASD cases in specific regions of Chr X. Specifically, we combined variants from SSC and SPARK and compared transmission of rare damaging variants in cases versus controls (Fisher exact test comparing rare maternally-inherited damaging variants versus rare non-inherited variants in cases versus controls). Because each window only had a tiny number of variants, very few windows were statistically significant, although the odds ratio did indicate a general enrichment trend in some windows. Windows with OR > 1.2 are highlighted in red (each dot denotes one window). All four RERs (red vertical shadows) overlap multiple windows of interest (OR>1.2). The third RER region only has a few windows, which may be because of the relatively low gene density in that region of Chr X. It is also interesting to note that this analysis identifies two additional regions that may also carry risk. Given the small number of “informative” families (n = 65) used to detect RERs and the low recombination rate in these two regions (due to their close proximity to the centromere), we hypothesize that we did not detect these regions in our main analysis due to a lack of power.

- The result of the authors that damaging variants are more inherited in RERs seems to me to be a bias from the selection of the RERs. By selecting regions that are inherited only in affected males any inherited LGDs will by definition become enriched in these regions.

We understand the reviewer's concerns about tautology and thank them for raising this feedback. However, we believe that this result is not tautological for three major reasons.

First, we defined these regions based on a subset of SSC families ($n = 65$). We chose to call them risk enriched regions or RERs based on the hypothesis that rare variants within these regions may carry male specific risk. To test this hypothesis, we assessed the rate of rare damaging variants within these regions in affected versus unaffected male probands. Importantly, we conducted the aforementioned burden analyses in much larger groups of samples (e.g. 1,014 male probands from the SSC; e.g. 11,391 male probands from SPARK). Thus, these samples are largely independent from the definition of RERs and therefore we believe that the enrichment of damaging variants within these regions indicates that our strategy successfully identified regions of Chr X containing genes that, when mutated, impart predominantly male risk (i.e., the test results supported our hypothesis above). However, to the reviewers point, we did include the 65 SSC families used for identification of the RERs in our burden analyses. Therefore, to rule out any bias that may have been introduced by using the same samples for both RER identification and burden analysis, we repeated the SSC burden analysis, excluding these 65 families. Enrichment of rare damaging RER variants did not appear to change after removing these samples (OR 1.59, $p = 0.011$ versus OR 1.60, $p = 0.0084$).

Second, the analyses in Figure S4B (discussed directly above) do not rely on selecting regions that are inherited only in affected males. Instead, we leveraged the combined SSC and SPARK cohort to look for regions where rare damaging variants are over transmitted to cases versus controls. This orthogonal method identified regions that strongly overlap with

the RERs defined based on informative recombinations in the 65 SSC families—providing additional validity to the definition of the RERs and suggesting our results are not driven by tautology.

Finally, the top 4 genes identified for ASD (Figure 4) do not have any variants from the 48 families used for RER identification.

- I'm concerned about the strength of the statistical evidence. Most of the significant results that the authors present barely reach significance. In addition, I note that the authors perform only single sided testing, whereas two-sided testing would be appropriate. In addition, the authors do not perform any multiple testing corrections when testing for example for multiple different categories of variants (missense, LGDs or recombination genes, overlap, RER genes). (Except for the TDT where authors perform bonferroni correction).

We chose to use one-sided testing for our burden analyses because more than a decade of whole-exome sequencing-based research in ASD (exploring both the autosomes and Chr X) has consistently demonstrated an excess of deleterious variants in cases versus controls. Additionally, given that our model was built on the foundation that male vulnerability is due, in part, to deleterious hemizygous variants on Chr X, we would independently hypothesize that damaging variants on Chr X would be increased in cases versus controls.

Regarding correction for multiple testing, our main goal was to determine whether rare damaging variants within specific regions of Chr X carry predominantly male risk; i.e. our primary hypothesis was that rare inherited damaging variants on Chr X within RERs would be enriched in male cases versus controls. Most of the additional burden tests (e.g. Chr-X-wide LGD and missense enrichment) were solely conducted for comparison with previously published studies, for validation of our results (e.g. assessment of common variants as a negative control, e.g. assessment of variants in female cases versus controls), or were exploratory analyses (e.g. assessment of highly brain expressed genes, e.g. comparison of high and low recombination genes). In the case of the exploratory analysis expanding to assess TS, ADHD, and EE, perhaps we should have corrected for multiple comparisons—however, these results would be unchanged after Bonferroni correction (TS adjusted p-value = 0.0001, ADHD adjusted p-value = 0.0001, EE adjusted p-value = 1). Therefore, we have updated the main text with Bonferroni corrected p-values for the association of rare damaging variants on Chr X with TS, ADHD, and EE.

- I note that strangely enough when the authors reproduce their results in a much larger SPARK cohort, the statistical significance does not really increase. This is not what I would expect. Can the authors provide some kind of explanation for this?

This is a great question and highlights that we should have been more explicit about this in the main text. For the discovery analysis in the SSC (n = 1,014 male probands, n = 746 unaffected male siblings), we performed a case-control burden analysis using a Fisher exact test (damaging versus synonymous variants in RERs in cases versus controls). However, for the validation analysis using the SPARK cohort (n = 11,391 male probands, n = 1,549 male sibling controls) we chose to use an orthogonal test so as to strengthen validation of our findings. Specifically, we performed a transmission disequilibrium test (TDT). As a result, it is

not meaningful to compare the two p-values directly. We apologize for the confusion and have clarified this in the main text.

Nonetheless, to further address this comment, we conducted the same TDT approach using the SSC samples only and observed that over transmission of rare damaging variants in RERs, but not NERs, trended towards significance (RERs: OR 1.39, $p = 0.12$; NERs: OR 0.78, $p = 0.99$). To better interpret this result, we assessed the statistical power of this analysis. Specifically, we performed simulations with a fixed case-to-control ratio based on the combined SSC & SPARK dataset, 13,052 cases and 2,295 controls) to get an estimate of the statistical power under different sample sizes. As shown below, the power to detect a nominally significant result is 17% in the SSC dataset, but is 64% in the SPARK dataset, and 70% in the combined dataset.

In addition, we carried out a case-control burden analysis for SPARK (i.e. using an analogous Fisher exact test for damaging versus synonymous variants in RERs in cases versus controls), which again revealed that rare damaging variants within RERs are significantly enriched in ASD cases (RERs: OR 1.25, $p = 0.045$; NERs: OR 1.02, $p = 0.36$), but with a smaller effect size and larger p-value than observed in the SSC dataset (OR 1.60, $p = 0.0084$ for RERs). The difference in effect sizes is not unique to our study. The effect size for autosomal de novo protein-truncating variants (variants that are robustly associated with ASD) identified in a recently published study (PMID: 35982160) varies substantially between the two cohorts (SSC: OR 1.79, $p = 2.93E-7$; SPARK: OR 1.37, $p = 3.39E-5$). This could reflect a general difference in the ascertainment of the cases and controls in the SSC versus SPARK.

Vice versa, for example, in the ADHD cohort. Although the authors cannot verify the transmissions status (thereby diluting their dataset) and the cohort is much smaller than the

ASD cohort, the author still find significant enrichment. In the ADHD cohort the enrichment could have been purely driven by de novo variants on chrX? Can the authors exclude this?

Since the frequency of de novo mutations is substantially smaller than that of transmitted variants, they should theoretically have little impact on the burden analysis. Nonetheless, we utilized SSC samples as an example to investigate the potential impact of de novo mutations (because we were unable to get maternal data for the ADHD cohort). We identified de novo mutations based on the variant being hemizygous in a male child and homozygous in the corresponding maternal data. We did not use any filters to attain high sensitivity to find de novo candidates. As a result, from 1014 probands and 746 siblings, we identified 9 and 15 putative de novo variants, respectively. After excluding all of these de novo variants, the enrichment of rare damaging variants within RERs remains (before vs after: OR 1.45, $p = 0.027$ vs OR 1.48, $p = 0.021$). According to a recent study (PMID: 31768057), the mutation rates in ASD and ADHD should be comparable, therefore we anticipate that de novo variants in ADHD should also be very rare and should not have a significant impact on the signal we saw in the ADHD cohort.

It is also worth mentioning that when “converting” the SSC burden tests from maternally-inherited to all rare hemizygous variants (i.e. not incorporating information about inheritance, and thus, potentially including de novo variants), the effect size does not shift markedly (OR 1.45, $p = 0.027$ for case-control versus OR 1.60, $p = 0.0084$ for maternally transmitted). This suggests that the ADHD and TS cohort estimates from all rare hemizygous variants should be similar to the estimates that would be obtained if we used parental data.

- The authors at several points draw a conclusion from obtaining non-significant results after testing which is invalid. For example the authors conclude that RER risk variants are not associated with NVIQ based on a non-significant p-value.

We agree with the reviewer that interpreting non-significant p-values is invalid. In the case of the NVIQ comparison, however, the null hypothesis is that there is no NVIQ difference in male probands with damaging RER variants compared to male probands without damaging RER variants. Since we observe a non-significant p-value, the null hypothesis cannot be rejected. Therefore, we think it is fair to state that RER risk variants do not appear to impact NVIQ. However, in line with the reviewer’s comment we have removed the sentence “In fact, the enrichment of rare damaging variants may be higher in the above average NVIQ group (OR 1.40 $p = 0.10$; Figure S2C),” as this statement is based on a non-significant p-value.

- The authors often test for effects in females, and from obtaining non-significant results claim support for a male-specific effect. However, the cohort of affected females is considerably smaller than for males and therefore relatively underpowered compared to the male cohort. Given the marginal statistical significance in males it is unlikely that significance can be obtained in the females even if the effect sizes are identical.

This is a great point and we appreciate the feedback. We agree that this is a limitation and this is why we have chosen to use language like “predominantly male risk” rather than “male-specific risk”. Regardless, we agree that this requires more consideration, and therefore, we have added this point to the discussion section, as part of a new paragraph on limitations of the study. We have also “softened” our conclusions about the lack of signal in the female

samples (e.g. changing “rare maternally transmitted heterozygous damaging variants in female probands are not enriched within the RERs” to “rare maternally transmitted heterozygous damaging variants in female probands do not appear to be enriched within the RERs.”

Minor comments:

- The brain expressed results, making use of the 50% cutoff is not that convincing. The results would be much more convincing if RER genes would be enriched in the top 10% or 5% expressed genes. What brain tissue was used here? Why do the authors compare to the full exome, rather than the RER genes, or RER genes with inherited LGDs? I did not find any information about this analysis in the methods.

A specific threshold of expression for relevance to the brain is difficult to define (e.g. in BrainSpan, ~80% of genes are considered to be brain expressed; PMID: 22031440). Therefore, we chose to compare Chr X genes in the top 50% of brain expressed genes to the remaining Chr X genes in order to conduct a well-powered analysis. However, to address the reviewer’s concerns, we repeated the analysis, comparing the Chr X genes in the top 25% of brain expressed to the remaining Chr X genes and observed consistent enrichment of rare damaging variants within highly brain expressed RER genes in ASD probands (OR 2.84, $p = 0.031$ for top 25% versus OR 2.10, $p = 0.015$ for top 50%). We have added this result to the main text. Comparing genes within the top quartile of brain expression to remainder of genes is a common heuristic in whole exome sequencing studies (e.g. PMID: 23665959, PMID: 26785492)

We utilized brain expression data from all of the male BrainSpan samples in order to rank the gene list. In this particular analysis, we did not focus on any particular regions of the brain or developmental stages in order to show the overall level of gene expression. In the gene discovery table (Table S1-3), we also included the gene expression level that we calculated from regions of the brain that we have previously highlighted (PMID: 24267886).

We ranked the genes exome-wide under the presumption that, regardless of genomic location, genes with a higher brain expression level are more likely to be functionally important in the brain in general. However, to address the reviewer’s comment we reran the analysis, ranking Chr X genes relative to other Chr X genes only. We observed very similar results for genes within RERs (Chr-X-wide top 50% brain expressed genes: OR 2.21, $p = 0.0080$; remaining genes: OR 1.50, $p = 0.086$ versus exome-wide top 50% brain expressed genes: OR 2.10, $p = 0.015$; remaining genes: OR 1.50, $p = 0.085$).

To clarify how we perform this analysis, we added more details to the method section under “Burden analysis regarding elevated ADHD symptoms, non-verbal IQs, brain-expression levels”

- In the TDT please show the corrected p-value for MAGEC3 as well. This is more informative to the reader in assessing the robustness of the association.

We thank the reviewer for the suggestion. We added Bonferroni adjusted p-values and Benjamini-Hochberg FDRs for both Chr-X-wide level and exome-wide level to the main text and to supplemental Table S5.

- The analysis that the authors claim shows that their results in RERs is not driven by recombination is not very clear or convincing to me. The analysis is based on the 149 genes with highest surrounding recombination rate, but this does not mean that genes outside of the top 149 do not still have a high recombination rate. Again, please include the underlying data that was used for the analysis in order to allow readers to verify the results.

We thank the reviewer for the comment and agree that we needed to delve more deeply into this issue. Because our strategy to identify RERs heavily relies on “informative” recombination events, our goal is to understand whether the identification of risk regions is driven solely by recombination rate. Therefore, our results shown in the original Figure 3 were meant to demonstrate that selecting genes purely based on recombination rate does not perform as well as our model. To address the reviewer’s concern, we conducted two additional analyses (shown below). First, we compared linear regressions to determine whether using recombination rate as a covariate impacts prediction of the number of damaging variants (Figure A). Our results suggest that the recombination rate appears to have no effect on the rate of rare damaging variant counts in RERs ($p = 0.24$, F-test), NERs ($p = 0.86$), and across all Chr X genes ($p = 0.62$). We added the Chr-X-wide result to main text Figure 3C, and the rest of these results to supplemental Figure S4. Second, we split the genes in RERs/NERs into high and low recombination rate groups (i.e. top 50% versus bottom 50%). We then conducted a burden analysis in each gene group (comparing the rate of rare damaging variants in male probands versus unaffected male siblings) and checked whether there appeared to be any difference between the effect sizes for genes with high versus low recombination rate (Breslow-Day test for homogeneity of the effect, Figure B). In both RERs and NERs, we did not observe sufficient evidence to reject the null hypothesis of homogeneity ($p = 0.24$ for RERs, $p = 0.74$ for NERs). These results have been added to supplemental Figure S4 as well. All together, these observations suggest that the genes in RERs carry risk somewhat independent of their recombination rate. We have included the datasets and scripts to reproduce this analysis on bitbucket: https://shengw@bitbucket.org/willseylab/chrX_analysis.git

- The association of MAGEC3 is based upon a single statistical analysis without independent replication or other supporting evidence. The authors should therefore be careful to suggest this gene as a high confidence risk gene.

We understand the reviewer's concerns. However, "high confidence" in rare variant / exome sequencing analyses of psychiatric conditions is typically defined based on exome-wide significance after Bonferroni correction ($p < 2.5E-6$) or based on a false discovery rate (FDR) less than 0.1, regardless of whether or not the association has been replicated (PMID: 35440779). In our case, MAGEC3 meets both of these conditions and so we feel that "high confidence" status is consistent with the field.

- As the authors point out there are many comorbidities in neurodevelopmental disorders. The authors take this into account by performing regression analysis taking into account comorbidity. This seems to be the right approach compared to the analysis mentioned before where the cohorts are analyzed without taking into account comorbidity status. I feel that the latter analysis should therefore be removed from the manuscript. Please also provide a supplementary sample overview with the phenotypes for the samples that were used.

With respect to the second part of this comment, we agree with the reviewer that these data should be made available. As mentioned previously, to allow readers to reproduce all the figures and tables in this study, we have now included full datasets along with the scripts on bitbucket: https://shengw@bitbucket.org/willseylab/chrx_analysis.git. This includes detailed phenotypic information.

With respect to the first part of this comment, we appreciate the reviewer's kind words about the regression analyses. However, we feel that both analyses are valuable. The cohort by cohort analyses (Figure 3A) provide useful information about the overall burden of rare Chr X variants expected in a population ascertained based on the index disorder (e.g. ASD). While there are of course ascertainment biases within each cohort, this information may nonetheless be useful to researchers and clinicians (e.g. to estimate the expected yield of risk variants). In contrast, the regression analyses taking into account comorbidity provide somewhat abstracted insight into the amount of risk that rare Chr X variants carry for each disorder and are therefore useful for understanding the relative contribution of these variants to risk.

Reviewer #2 (Remarks to the Author):

In this thoughtful and extensive set of analyses the authors probe the role of X-chromosome variation in male-biased neurodevelopmental disorders. In short, they used family-based ASD data to identify risk enriched regions in the X chromosome that roughly correspond to recombination hotspots. They observed that maternally inherited rare "damaging" variants within these regions are enriched in male ASD cases. Furthermore, these variants are also enriched in males with Tourette's Syndrome and ADHD (common comorbidities that are similarly male biased) but not in males with epileptic encephalopathies (a common comorbidity that is not male biased). Overall, this adds important insights into the influence of X-chromosome variation on disease and the likely etiologic overlap between several neurodevelopmental conditions. A few points deserve consideration, and may further improve this work.

We thank the reviewer for these kind comments and helpful feedback.

1) Line 171 and Figure S2: Odds Ratios are presented but their calculation and interpretation in words could use some clarification. The odds of [what?] are higher in male probands. The concept of enrichment is raised, but it could help the reader if the Odds Ratios throughout the manuscript were explicitly clarified in this way.

We apologize for the confusion. We estimated odds ratios (ORs) using a Fisher exact test, generally comparing the rate of damaging versus synonymous variants in cases versus controls. We included synonymous variants in the contingency table in order to correct for ancestry and/or batch effects (e.g. differences in variant calling due to differences in the underlying exome array; see Figure S1, PMID: 30257206). Practically therefore, an OR > 1 means that cases are more likely than controls to carry a rare damaging variant, accounting for batch effects (e.g. if we saw 10 damaging variants and 20 synonymous variants in cases, and 5 damaging variants and 20 synonymous variants in controls, the OR would be 2, meaning cases are 2x as likely to have a damaging variant as compared to controls [i.e. the rate of damaging variants is 2x higher in cases]).

Within the “Burden analysis” section of the methods we have added a new subsection as well as more details to existing subsections in order to make this more clear. We have also clarified the details of the odds ratio calculations in the main text as needed.

2) It may help to note the difficulties with variant pathogenicity assessments, especially regarding the lack of a gold standard. The authors should not have to redo the analyses, but a threshold from an Ensemble method such as REVEL may have been a better choice for defining “damaging” variants. <https://pubmed.ncbi.nlm.nih.gov/32843488/> A brief comment on this topic could further improve the manuscript.

We thank the reviewer for this suggestion and agree that it is challenging to annotate damaging variants, particularly missense variants that may result in either function gain or loss. In previous studies, it has been shown that variants with higher PolyPhen-2 HDIV scores—predicted as “possibly damaging”—are significantly associated with ASD risk. Therefore, we used the same annotation for the analysis in this study.

We also tried REVEL as suggested. The cutoff set by the reference work (PMID: 32843488, REVEL score > 0.5) was not applicable in our situation, as this was generated mostly based on autosomal variants. Thus, we annotated REVEL scores for pathogenic/likely pathogenic variants from ClinVar (11 X-linked recessive disorders; 70 variants), and rare variants from non-neuro GnomAD samples (which are assumably neutral). We further used all REVEL scores on Chr X as a control. From the proportion distribution below, we employed a stricter cutoff (REVEL score > 0.6) to more precisely predict damaging variants. Even though the number of variants in our study passing this cutoff was extremely small, RERs alone still trended towards enrichment (OR 1.53, $p = 0.16$ for REVEL variants compared with OR 1.61, $p = 0.0078$ for Missense 3 variants). Consistent with this result, 81% of missense variants with REVEL score > 0.6 are classified as Missense 3 and were therefore considered as damaging variants in our study. Conversely though, only 22% of Missense 3 variants would be retained if using the more stringent REVEL cutoff, greatly reducing power for our study (likely resulting in the non-significant p-value observed above for REVEL variants).

Regardless, we agree this is an important point, especially in the context of Chr X mutations (which may require different predictors and would likely benefit from optimization), and therefore, we have added a brief discussion of this point to a new “Limitations” section in the Discussion.

3) Lines 185 to 194: This is an interesting idea and there are exiting directions that it could take. However, the analysis as currently described does not warrant the conclusion offered. If the authors are interested in RER risk variants as a potential cause of low NVIQ, then they should probably not look among probands only (ASD cases). NVIQ may be a partial determinant of proband identification. Thus, in this setting, conditioning on ASD status is conditioning an intermediate variable, or a putative effect of exposure, or a proxy for the outcome under study, or (more simply) a likely collider. This would be expected to generate bias. Thus, it probably makes more sense to look for these associations in all participants, or even better, in another population-based dataset where inclusion was not influenced by ASD status. Interestingly, the presence of an NVIQ-VIQ discrepancy may define an important ASD endophenotype or identify a general category of aberrant neurodevelopment that could lead to several diagnoses <https://pubmed.ncbi.nlm.nih.gov/24450323/> The authors could consider exploring the enrichment of RER risk variants within this group (i.e. NVIQ > VIQ - agnostic of ASD status).

These are great points and we agree that it would be better to conduct such an analysis in a different dataset using more sophisticated approaches. However, our goal here was simply to test whether damaging RER variants have large effects on NVIQ. This is an important question for two reasons. First, de novo damaging variants on the autosomes are associated with decrements in NVIQ (e.g. see PMID: 26402605, PMID: 31981491). Second, many intellectual disability (ID) genes are X-linked (PMID: 33504798) and there is much debate in the field about whether high confidence ASD genes carry “ASD-specific” risk or “share” risk with ID (PMID: 35440779). Thus, an absence of a large effect on NVIQ in autism probands (as we observed) would suggest that the genes identified by damaging RER variants are not simply “ID” genes as well as support the idea that damaging RER variants appear to have less impact on NVIQ than de novo autosomal variants. The latter is quite interesting given the similar effect size of damaging RER variants on Chr X and de novo autosomal variants.

In response to this question, we have simplified the analyses to a primary comparison of NVIQ between probands with and without rare damaging variants in RER genes (Figure S2C) and use more cautious language when interpreting the result. We have also added comparisons of verbal IQ (VIQ) and full-scale IQ. We have added a few sentences to the discussion elaborating on this point as well.

4) Line 232: Why are one sided tests used in some instances? This is rarely an appropriate choice unless it is explicitly defended in the text.

We chose to use one-sided testing for our burden analyses because more than a decade of whole-exome sequencing-based research in ASD (exploring both the autosomes and Chr X) has consistently demonstrated an excess of deleterious variants in cases versus controls. Additionally, given that our model was built on the foundation that male vulnerability is due, in part, to deleterious hemizygous variants on Chr X, we would independently hypothesize that damaging variants on Chr X would be increased in cases versus controls. Similarly, for the enrichment tests of the RER genes versus recombination genes versus the union of these genes, the test was based on a clear hypothesis that these gene sets would be enriched for damaging variants within RER genes.

5) Line 257 Bonferroni correction is quite conservative. It would be interesting and perhaps important to see how many genes had a significant over-transmission with a less conservative approach (e.g. FDR). The trade off between type 1 and type 2 error is particularly important in discovery stage research <https://pubmed.ncbi.nlm.nih.gov/25071867/> and these results could be presented in the supplement.

We thank the reviewer for the suggestion. We added both Bonferroni corrected p values and Benjamini-Hochberg FDRs in the new supplemental Table S5.

6) The authors appear to have further exposed the importance of deleterious hemizygous variants in male-biased Neurodevelopmental disorders. Brief mention of recent work on hemizyosity-revealed pathogenicity <https://pubmed.ncbi.nlm.nih.gov/35994124/> and long standing medical genetic observations <https://pubmed.ncbi.nlm.nih.gov/15316978/> <https://pubmed.ncbi.nlm.nih.gov/17707844/> may further improve the discussion.

We thank the reviewer for the suggestion. We now mention these in the main text and/or discussion.

Minor Points

Note that the introduction reads like a summary of the whole manuscript (including results and discussion).

Table 1 legend: “can be explained by” is strong wording that invokes causal interpretations. This probably should be tempered (“may be explained by” could work better here).

Agreed. We have changed the section in the legend to “Overall, we estimate that more than 2% of male cases could be explained by rare damaging variants in RERs and that approximately 20% of damaging variants carry risk.”.

The paragraph on line 51-55 could use citations (e.g. <https://pubmed.ncbi.nlm.nih.gov/32284538/> , <https://pubmed.ncbi.nlm.nih.gov/15316978/> , <https://pubmed.ncbi.nlm.nih.gov/35994124/>)

We added these references in the revised manuscript.

Reviewer #3 (Remarks to the Author):

In this well-written manuscript, the authors examined the role of inherited coding variation in Autism Spectrum Disorder. They targeted rare, maternally inherited regions and variants as a possible contributor for the male-bias in ASD. The authors presented results that were inline with the literature and they attempted to implicate new risk genes; they described a new link between the X-linked gene MAGEC3 and ASD. They used data from ADHD/TS - traits with male-bias - to test whether the contribution of inherited variants is pertinent to male-specific risk in ASD-correlated traits, utilising DEE - a disease without sex bias - as a negative control set. The methods are appropriate and their description is adequate. The interpretation of the findings is largely well-justified, albeit with occasional extrapolations based on non-significant findings. I would suggest a more conservative wording when it comes to statements made on male-specific risk or comparisons where merely the point estimates of odds are different. I hope the authors find the specific feedback below useful to their revision.

We thank the reviewer for these kind comments and helpful feedback.

1. It was found that the RER genes are enriched for those with a high recombination rate - although they have a broader distribution. A reference, equally sized, set composed of the 149 genes with the highest surrounding recombination rates in HapMap was used for comparison and the difference in distribution was interpreted to suggest that the two gene sets may have different patterns of risk.

All in all, It is rather difficult to argue in either direction and the statements "rare damaging variants are significantly more likely to occur within the 99 RER-only genes than within the 99 recombination-only genes" in the results and "most of the risk for rare transmitted damaging variants resides within RER genes" in Fig. 3 legend are too strong for several reasons, for instance:

- Both 'unique' sets are not significantly enriched and the signal is driven by the shared genes (at least based on p values). The p value for the RER-only genes is 0.85 and for the Recomb-only genes is 0.38; the CI for the odds are not given and point estimates of the odds on their own, when the p values are this far from the lower end, are not very informative.
- The 99 recombination-only genes are by definition a subset of non-RER. The whole premise is that non-RER damaging variants are not as enriched as RER variants. Directly comparing RER vs. Top-ranked on Recombination Rate should therefore not be considered an 'additional layer of evidence' .e.g, in "Together, this suggests that the majority of risk is present within the RER gene set".

- It would appear that the RER genes resemble the general distribution of recombination rates along chrX more than the ranked set. This diminishes the suitability of 'top-ranked' as a fair comparison set.

Would it be possible to approach this differently to account for the independence? Some naive suggestions (i) by using the recombination rate as denominator for normalisation as the authors did with synonymous variants, (ii) using a regression model with the recombination rate as a covariate or (iii) exact tests between high/low recombination while using RER/non-RER as strata (or the other way around) then testing for homogeneity of effects along the two strata.

We thank the reviewer for the kind suggestions. It is difficult to do a normalization using recombination rate as a denominator as was done with synonymous variants, counts of which are significantly associated with rare damaging variants (Pearson's $r = 0.36$, $p < 2.2E-16$). Therefore, we performed the other two analyses to check whether the RER identification is independent of recombination. (1) Fig A, we compared the linear regression models for variants within RERs/NERs with the formula $\#ssc_pro.Dam \sim \#ssc_sib.Dam + \log(\text{recombination rate})$ and $\#ssc_pro.Dam \sim \#ssc_sib.Dam$. We transformed the recombination rate to a log scale in order to make its distribution more normal. We then performed F-tests to compare the two models and observed no significant difference ($p = 0.62$ for all Chr X genes, $p = 0.24$ for RERs, and $p = 0.86$ for NERs). These results therefore suggest that recombination rate is not a significant predictor of the count of rare damaging variants in probands. We added the Chr-X-wide result to main text Figure 3C, and the rest of these results to supplemental Figure S4. (2) Fig B, we split the genes within RERs or NERs into high and low recombination groups and tested the homogeneity of the enrichment of rare damaging variants in ASD probands. To do this, we ranked the genes on Chr X NONPAR based on recombination rates and split the genes into "high" and "low" recombination bins (50-50 split). We did not observe a significant difference in the effects between genes with high and low recombination rate in both RERs and NERs (Breslow-Day test $p = 0.24$ for RERs and $p = 0.74$ for NERs). That being said, the RER genes with a lower recombination appear to trend towards being more enriched, though this is not a conclusive result. Altogether, these results support the assertion that recombination rate does not appear to be a primary driver of the risk carried within RERs. These results have been added to supplemental Figure S4 as well.

The authors also touched upon the role of expression levels of RER-genes as a plausible covariate that could modulate the enrichment. It could be relevant to see the outcomes of a control comparison in which the enrichment is examined when the genes are stratified by brain expression, agnostic to their location in RER/non-RER.

As per the reviewer's suggestion, we checked the burden of damaging variants within genes grouped based on brain-expression levels alone: we did not observe enrichment of rare damaging variants within the top 50% of brain-expressed genes (OR 1.08, $p = 0.30$) nor the bottom 50% of brain-expressed genes (OR 0.86, $p = 0.92$).

2. IQ is yet another important covariate the the authors addressed in their analysis. For verbal IQ comparisons, the cohort was split into 'above average' and 'below average' which is an unusual approach; IQ is usually dichotomised using cut-offs that would separate the lower tail (Well below average or extremely low IQ) rather than a "50/50" split around the mean/median.

- Is there any explanation for this choice and are the outcomes comparable for 'more familiar IQ group definitions'?

- How much evidence do the authors find for the statement "the enrichment of rare damaging variants may be higher in the above average NVIQ group; OR 1.40 $p = 0.10$ ", in light of the literature?

Here, I would use more conservative wording as a significant difference is far from noticeable; Although not given in the text, I expect the 95% confidence intervals are too large with a p value of 0.1 to justify a possibility of a higher burden in the high IQ group.

We thank the reviewer for raising this important point and we agree that our cut-offs were unusual. Our goal here was simply to test whether damaging RER variants have large effects on NVIQ. This is an important question for two reasons. First, de novo damaging variants on the autosomes are associated with decrements in NVIQ (e.g. see PMID: 26402605, PMID:

31981491). Second, many intellectual disability (ID) genes are X-linked (PMID: 33504798) and there is much debate in the field about whether high confidence ASD genes carry “ASD-specific” risk or “share” risk with ID (PMID: 35440779). Thus, an absence of a large effect on NVIQ (as we observed) would suggest that the genes identified by damaging RER variants are not simply “ID” genes as well as support the idea that damaging RER variants appear to have less impact on NVIQ than de novo autosomal variants. The latter is quite interesting given the similar effect size of damaging RER variants on Chr X and de novo autosomal variants. However, in response to this comment, we have removed the analysis where we group individuals into “high” and “low” NVIQ and instead compare the distribution of NVIQ between probands with a damaging rare variant in an RER gene to those without. Additionally, we added a comparison of verbal IQ (VIQ) and full-scale IQ. We also use more cautious language when interpreting the result and have added a few sentences to the discussion elaborating on this point.

3. A very interesting part of the analysis was the comparison between TS/ADHD/DEE sets and SSC male siblings. Once again, I would push back a bit on the strength and the generalizability of the statement "All together, this suggests that rare damaging variants within RERs predominantly carry risk for males in male-biased NDDs only".

- A disease without a prominent male bias AND a considerable role for inherited variants is needed to examine the premise that inherited variants contribute to male-bias. While indeed a reasonable control when it comes to the phenotypic male-bias, DEE is a poor control when it comes to the mode of inheritance. DEE is caused predominantly by de novo variants and inherited variants are a very unusual cause of DEE unless bi-allelic.

We thank the reviewer for the helpful comment. We would, however, like to note that a substantial contribution to risk from de novo variants is true in many neurodevelopmental disorders, such as ASD and TS, particularly for simplex families where both parents are unaffected (like the SSC samples). In support of this, a recent study (PMID: 31981491) has shown that the liability of de novo protein-truncating variants (PTVs) in ASD is far greater than that of transmitted PTVs. Moreover, rare damaging variants within male Chr X non-PAR are hemizygous, which should, in theory, have the same penetrance as bi-allelic variants on autosomes.

Regardless, we agree with the reviewer that assessing a single, relatively small cohort is not strong evidence to support the statement that rare damaging variants within RERs carry predominantly male risk in male-biased NDDs only. Therefore, we conducted an additional analysis using published data for a much larger cohort of individuals (7,136 male cases, 8,551 male controls [unaffected fathers]) with severe, undiagnosed developmental disorders (DD) and a limited male-sex bias. Specifically, we leveraged rare Chr X non-PAR variants from a recent Deciphering Developmental Disorders study (PMID: 33504798). Their cohort had a sex bias of about 1.4 (male:female), which is much lower than what is typically observed for ASD, TS, and ADHD. Hence, we postulated that the enrichment of rare damaging variants in DD should be lower than that in ASD, TS, and ADHD. We obtained the variant-level information from their manuscript (https://github.com/hilarymartin/DDD_chrX.git) and annotated variants with the same pipeline as we used in this study. With an analogous burden analysis, we observed that rare damaging variants in DD male cases are not significantly enriched in RER genes (OR 1.09, $p = 0.10$).

- Analysing a very small set of DEE cases against a set of SSC siblings - on top of the low power - creates further ascertainment bias as several genes cause ASD/DEE/NDD-Epilepsy and limiting the analysis to RER not segregating in siblings enriches the control set for individuals without variants more than what would have been expected under a null model.

We agree completely that the EE analysis is underpowered. However, the new DD analysis described above is very well powered and therefore, we hope it mitigates some of your concerns.

We also agree that epilepsy and ASD are commonly comorbid and/or share risk genes (e.g. *SCN2A*), so it is very reasonable that there are shared genetic risks between the two conditions. However, the shared risk is not necessarily located inside the RERs, and this is in fact part of what we are testing.

In addition, to clarify, the vast majority of the samples included in the burden analysis were not used in RER identification and exclusion of these overlapping samples ($n = 65$ families) does not affect the results: OR 1.59, $p = 0.0108$ when removing the 65 families versus OR 1.60, $p = 0.0084$ when including these families. Therefore, it is not guaranteed that rare inherited variants would be enriched within these regions as a matter of course because the vast majority of analyzed samples were not used in RER identification. In other words, identification of the RERs in a small number of families should not be influencing the number and type of variants segregating in RERs in the rest of the families and/or cohorts utilized. In support of this idea, in ASD we observe significant enrichment of rare damaging variants within these regions but not rare synonymous variants.

Finally, the top 4 genes identified for ASD (Figure 4) do not have any variants from the 48 families used for RER identification.

4. Although the argument was made in a technical context, the authors suggest the validity of case-control analyses as direct comparator of effect sizes. I wonder whether the authors could discuss briefly the reasons behind the absence of *MAGEC3* from the top-hits of case-control studies that analysed similar SCC and SPARK cohorts. My impression is, it is fairly unusual for high-confidence genes to change rank too much between studies using largely overlapping sets. A little more elaboration on the biological plausibility of *MAGEC3* would also be very useful. The role of the MAGE family in NDD was briefly discussed. It seems there are four paralogs of MAGE among the top hits of the analysis (*MAGEC3*, *MAGEC1*, *MAGEE1*, *MAGED2* are among the top 10 genes). Would that be a fair expectation from the gene family size or do the authors believe there is possibly an enrichment in the whole gene set?

This is a great point. We think that *MAGEC3* is missing from the top hits of case-control studies analyzing similar SSC and SPARK cohorts for several reasons. First, in general, common and rare variation on Chr X tends to be somewhat ignored. For example, in the largest omnibus analysis of ASD conducted to date (which includes SSC and SPARK samples), Chr X and Chr Y rare variants were not included in gene discovery (PMID: 35982160). Second, we observed systematic, substantial undercalling of rare maternally-inherited variants on Chr X, and therefore, designed a modified transmission disequilibrium test to account for this systematic bias (which acts in a direction to reduce power). Other case-control studies of Chr X in ASD may not have corrected for this systematic undercalling when conducting a TDT. We have added text to the discussion elaborating on this.

The link between disruptions of *MAGEC3* and ASD is unknown. However, MAGE family proteins regulate ubiquitylation, which has been linked with NDDs (PMID: 28300603). Interestingly, *MAGEC1* also has some evidence for association with ASD in our study, though it did not pass Chr-X-wide significance after Bonferroni correction ($p = 0.09$). This gene was also highlighted in a recent large-scale meta-analysis of schizophrenia whole exome sequencing data (PMID: 35396579), which is notable given the genetic overlap between these disorders. However, it is unclear whether MAGE paralogs are enriched within our top hits as we are underpowered to conduct this analysis. That being said, MAGE genes on Chr X tend to be included in RERs (12 of 34 MAGE genes are in RERs versus 149 total RER genes across 808 Chr X non-PAR genes, two-sided binomial $p = 0.023$). We now describe some of these points in the discussion.

5. The estimated percentage of cases in which RER variants contribute to risk is found to be around 2-3%, that is about 20-30 probands in the total set analysed for rare variant enrichment which included about a thousand male probands. Is there an overlap between the families used in discovering RER ($n = 65$) and the analysis set for rare variant enrichment (1k)? If that is indeed the case, could the enrichment be driven solely by this subset of (discovery) cases/families? Did the authors attempt to remove these families from their downstream analysis and see if the associations/enrichment would still be significant?

To clarify, 2-3% is not the total fraction of ASD probands carrying damaging variants within RERs, but the percentage of male ASD cases that may be explained by these variants. To arrive at this number, we estimated the difference between the rare damaging variant rate within RERs in cases and control samples (a common calculation in the field). Thus, 2-3% is the "extra" fraction in cases and therefore the proportion that may be contributing risk. In total however, around 10% of the ASD male probands, or roughly 100 probands in the SSC cohort, carry at least one rare damaging variant on RER genes.

To identify the RERs, we started with 65 families, but eventually narrowed to 48 families with informative patterns of Chr X segregation. We leveraged these 48 families to define RERs/NERs (Figure S2A). To confirm that inclusion of these 48 families was not confounding our analyses, we re-ran our burden tests after excluding these samples. The burden of rare damaging variants within RERs is virtually identical, whether these 48 families are included or not (OR 1.60, $p = 0.0084$ versus OR 1.59, $p = 0.0114$). We also obtain consistent results even when removing all 65 families (OR 1.59, $p = 0.0108$). Finally, none of our top genes in the TDT test (Figure 4) contain variants from these 48 families. We no mention these points in the main text.

6. It was clear described that the gene level case-control comparisons underwent FWER correction but I cannot clearly appreciate if the p values from the various other comparisons for enrichment analysis in the study were corrected. A lot of p values are close to 0.05 and it would be quite relevant to highlight if FDR/FWER adjustment was used, or the justification for not using any. It would also be very informative if the authors would consider adding confidence intervals to point estimates in the manuscript and to the QQ plots in Fig. 4 (e.g., using permutations/label shuffles or just an estimate from a drawn from a beta distribution). The genomic inflation parameters are odd; e.g., that the lambda from the non-RER QQ plot is lower than the RER QQ plot although the observed p values look closer to the expected ones

based on the drawn line; the 'overall' appearance of the RER plot is that of a slightly inflated one but lambda is less than 1.

We thank the reviewer for the suggestions.

First off, regarding correction for multiple testing, our main goal was to determine whether rare damaging variants within specific regions of Chr X carry predominantly male risk; i.e. our primary hypothesis was that rare inherited damaging variants on Chr X within RERs would be enriched in male cases versus controls. Most of the additional burden tests (e.g. Chr-X-wide LGD and missense enrichment) were solely conducted for comparison with previously published studies, for validation of our results (e.g. assessment of common variants as a negative control, e.g. assessment of variants in female cases versus controls), or were exploratory analyses (e.g. assessment of highly brain expressed genes, e.g. comparison of high and low recombination genes). In the case of the exploratory analysis expanding to assess TS, ADHD, and EE, perhaps we should have corrected for multiple comparisons—however, these results would be unchanged after Bonferroni correction (TS adjusted p-value = 0.0001, ADHD adjusted p-value = 0.0001, EE adjusted p-value = 1). Therefore, we have updated the main text with Bonferroni corrected p-values for the association of rare damaging variants on Chr X with TS, ADHD, and EE.

Regarding adding confidence intervals to the main text, we agree this would be helpful and so we have added these where appropriate.

For the QQ plots in Figure 4, we previously evaluated the lambda values at the median, but after reading your comment we realize that this is not that informative, especially when the number of genes included in the analysis is small. Following your suggestion, we have therefore removed the lambdas from the figure and added 95% confidence intervals (see below and also the new Main Text Figure 4).

7. Figure S4 suggests a trend of increasing effect sizes with lower MAF. With a fairly high MAF cut-off like 0.1%, it would be informative to comment on the actual MAF bin that seems to drive the enrichment (if these are mostly ultra-rare variants seen in one family), for example, in MAGEC3.

We are thankful for the reviewer's suggestion. We re-conducted the analysis with different MAF bins (<0.01%, 0.01-0.1%, 0.1 - 1%, 1 - 10%) and created the figure below (now Figure S5). Ultra-rare variants with MAF < 0.01% seem to be the key variants driving enrichment.

8. I would appreciate it if the authors elaborated a little in their discussion on the bias created by these two elements when it comes to inferring male-specific risk:

- (i) the design that scans males only for risk region and, to a lesser extent,
- (ii) the prioritisation strategy in families with more than one individuals.

We agree that these points warrant further discussion and have therefore added text addressing these points as part of a larger limitations section in the discussion. Thank you.

The authors set out to find, and successfully identified, regions within Chr X non-PAR that consistently segregated with risk in males. The design was made to prioritise regions carrying variation that have a preferentially high effect size in males (not in two unaffected males) but not as much an effect in females (inherited from a phenotypically unaffected mother).

Given that these regions were selected to segregate with ASD in males, the odds of an affected male having a rare variant in these region when compared to other (non-segregating) regions in affected males or to the same regions in (male) controls would anyways have a higher prior to begin with.

The exact tests used (with 'greater' as the alternative) assume the null is an 'equal' or a 'lower' burden. These should work just fine for gene-level scans without prioritisation on regions but will not capture the actual "null" distribution if limited to, or stratified by, segregating regions. It does not necessarily come as a surprise that inherited missense variants and PTVs in these regions are significantly enriched in males. The p values for variant enrichment could be nominally significant (see FWER above) and the enrichment could be a signal driven by the underlying haplotype sharing not the rare variants.

These are interesting points and we appreciate the reviewer raising them for clarification.

To begin with, we agree with the reviewer that RERs were identified based on patterns of segregation in males affected by ASD versus their unaffected male siblings. As the reviewer implicitly states, this was based on the hypothesis that these regions would carry male specific risk, and therefore, that (1) the odds of an affected male having a rare damaging variant within an RER would be higher than the odds of them having a rare damaging variant within an NER; and (2) the odds of an affected male having a rare damaging variant within an RER would be higher than the odds for an unaffected male.

However, we disagree with the idea that rare damaging variants would be biased towards occurring within these regions for several reasons.

First, we identified RERs based on informative recombination pinpointed via genotyping data (common variation) in a small number of families (48 after QC) yet conducted the initial burden analyses in over 1,000 samples. Therefore, it is not guaranteed that rare inherited variants would be enriched within these regions as a matter of course because the vast majority of analyzed samples were not used in RER identification. In other words, we could have identified these regions in and then observed that they are not enriched for rare damaging variants in probands versus siblings, because, for example, we solely identified the "null" recombination pattern for Chr X. The very fact that we see significant enrichment of rare

damaging variants within these regions (but not rare synonymous variants which would be similarly affected by such a bias if it existed) is precisely what suggests that they carry risk for ASD—especially because the vast majority of samples analyzed were not used in the identification of RERs and the signal remains even when excluding the samples used to identify RERs (the burden of rare damaging variants within RERs is virtually identical, whether these 48 families are included or not OR 1.59, $p = 0.0114$ versus OR 1.60, $p = 0.0084$).

Second, regarding the concerns of haplotype sharing, we checked the relatedness across samples before any analysis, and confirmed that the SSC samples are derived from different families—thus, it's very unlikely that there would be significant haplotype sharing between the 48 families used to identify RERs and the rest of the families analyzed. Moreover, for the families with both ASD probands and SSC unaffected siblings, we trimmed either probands (for males) or siblings (for females) to make sure the shared haplotype from the same family would not confound our analyses. Finally, we do not observe enrichment of common variants within RERs (or rare synonymous variants), further indicating that the enrichment of rare damaging variants is not driven by potential shared haplotypes in probands.

Finally, the top 4 genes identified for ASD (Figure 4) do not have any variants from the 48 families used for RER identification.

REVIEWERS' COMMENTS

Reviewer #1 (Remarks to the Author):

The authors have substantially revised their manuscript and have addressed many concerns by performing additional analyses that strengthen their conclusions. I have no remaining major concerns.

Reviewer #2 (Remarks to the Author):

Thank you to the authors for their thoughtful responses to my comments. An additional observation:

FDR is generally less conservative than Bonferroni, so more leads could in theory be identified if FDR adjusted p values are used. The FDR adjusted p-values are not necessarily required, but since they are now presented in the supplement . . . it appears that that Bonferroni and FDR-adjusted p values are identical. Please check these values. Thank you again for your thoughtful and extensive analyses.

Reviewer #3 (Remarks to the Author):

Excellent work!

REVIEWERS' COMMENTS

Reviewer #1 (Remarks to the Author):

The authors have substantially revised their manuscript and have addressed many concerns by performing additional analyses that strengthen their conclusions. I have no remaining major concerns.

We thank the reviewer for the positive comments.

Reviewer #2 (Remarks to the Author):

Thank you to the authors for their thoughtful responses to my comments. An additional observation:

FDR is generally less conservative than Bonferroni, so more leads could in theory be identified if FDR adjusted p values are used. The FDR adjusted p-values are not necessarily required, but since they are now presented in the supplement . . . it appears that that Bonferroni and FDR-adjusted p values are identical. Please check these values. Thank you again for your thoughtful and extensive analyses.

We are grateful for the reviewer's comments. We apologize for the mis-presentation of the FDR-adjusted p values, which we have now corrected and present in main text Table 2 and in Supplementary Data 1. Thank you again for pointing this out.

Reviewer #3 (Remarks to the Author):

Excellent work!

We thank the reviewer for the kind remarks of the revised manuscript.